# TerraFM: A Scalable Foundation Model for Unified Multisensor Earth Observation

Muhammad Sohail Danish[1]    Muhammad Akhtar Munir[1]    Syed Roshaan Ali Shah[2]
Muhammad Haris Khan[1]    Rao Muhammad Anwer[1,3]    Jorma Laaksonen[3]
Fahad Shahbaz Khan[1,4]    Salman Khan[1,5]
[1]Mohamed bin Zayed University of Artificial Intelligence    [2]University College London
[3]Aalto University    [4]Linköping University, Sweden    [5]Australian National University

## Abstract

Modern earth observation (EO) increasingly leverages deep learning to harness the scale and diversity of satellite imagery across sensors and regions. While recent foundation models have demonstrated promising generalization across EO tasks, many remain limited by the scale, geographical coverage, and spectral diversity of their training data, factors critical for learning globally transferable representations. In this work, we introduce **TerraFM**, a scalable self-supervised learning model that leverages globally distributed Sentinel-1 and Sentinel-2 imagery, combined with large spatial tiles and land-cover aware sampling to enrich spatial and semantic coverage. By treating sensing modalities as natural augmentations in our self-supervised approach, we unify radar and optical inputs via modality-specific patch embeddings and adaptive cross-attention fusion. Our training strategy integrates local-global contrastive learning and introduces a dual-centering mechanism that incorporates class-frequency-aware regularization to address long-tailed distributions in land cover. TerraFM achieves strong generalization on both classification and segmentation tasks, outperforming prior models on GEO-Bench and Copernicus-Bench. Our code and pretrained models are publicly available at https://github.com/mbzuai-oryx/TerraFM.

## 1 Introduction

Earth Observation (EO) provides systematic measurements of the surface of earth, supporting a wide spectrum of critical applications such as land use monitoring (Wang et al., 2023), crop evaluation (Prodhan et al., 2021; Kussul et al., 2017), urban development (Yu & Fang, 2023), and disaster response (Huot et al., 2022; Sarkar et al., 2023; Rahnemoonfar et al., 2021). These capabilities are enabled by a growing fleet of EO satellites, most notably the Sentinel missions, which deliver multi-modal, multi-temporal data at a global scale (Torres et al., 2012; Drusch et al., 2012). The rise of deep learning, particularly deep neural networks (DNNs), has fundamentally reshaped the way EO data is processed and interpreted (Wang et al., 2025; Szwarcman et al., 2024; Tseng et al., 2025). Modern DNNs enable automated extraction of spatial and semantic patterns from raw imagery, driving downstream tasks such as scene classification, object detection, and semantic segmentation (Astruc et al., 2024; Waldmann et al., 2025; Kuckreja et al., 2024; Tseng et al., 2025; Fuller et al., 2023). These models offer a scalable alternative to traditional hand-engineered pipelines by learning generalizable representations directly from data (Guo et al., 2024a). As EO data continue to expand in scale, diversity, and complexity, DNNs have become foundation for building high-capacity models capable of generalizing across geographies, modalities, and tasks (Szwarcman et al., 2024; Astruc et al., 2024; Waldmann et al., 2025).

Remote sensing data is inherently multimodal, comprising diverse sensor types such as optical, SAR, and multispectral imagery. Traditional EO pipelines often focus on single-modality inputs, typically high-resolution optical imagery, limiting the model's ability to generalize across varying sensing conditions. In contrast, multimodal and multispectral data sources, such as Sentinel-1 SAR and Sentinel-2 Level-1C/Level-2A optical bands, capture complementary structural and spectral information, enabling richer scene understanding (Han et al., 2024; Fuller et al., 2023). Foundation models that embrace this diversity have demonstrated superior transferability across tasks and

geographies (Tseng et al., 2025; Guo et al., 2024a). However, variation in ground sampling distance (GSD) across EO data makes tile size a critical factor; smaller tiles capture local detail but risk overfitting to texture, while larger tiles provide broader semantic context but require scale-robust architectures (Reed et al., 2023). Recent works like AnySat (Astruc et al., 2024) and msGFM (Han et al., 2024) have shown that scale-invariant modeling and mixed-resolution pretraining lead to more robust and generalizable representations. Crucially, large-scale sampling across geographies and resolutions enables EO foundation models to learn invariant features across sensors and global conditions.

As EO foundation models scale to accommodate diverse sensor inputs and resolutions, two dominant pretraining paradigms have emerged: masked autoencoders (MAE) and contrastive learning. Although MAEs focus on reconstructing the spatial structure, their reliance on RGB-centric ViTs limits their adaptability to multispectral or SAR inputs with varying spectral dimensions (Li et al., 2024; Szwarcman et al., 2024). In contrast, contrastive approaches such as DINO (Caron et al., 2021; Oquab et al., 2023) and its adaptations to remote sensing (Tseng et al., 2025; Fuller et al., 2023; Waldmann et al., 2025) offer modality-agnostic training by aligning global and local views through student-teacher distillation. However, the expansive spatial coverage of EO datasets introduces new challenges: large portions of satellite imagery are semantically sparse or uninformative, and naïve sampling can lead to representation bias. This requires intelligent sampling that prioritizes semantically diverse regions, guided by land cover priors, for balanced and efficient representation learning.

To address these limitations in standard ViTs, particularly their RGB-centric design, lack of modality awareness, and unimodal self-supervision, we introduce **TerraFM**, a unified foundation model tailored for remote sensing. First, we propose a *Modality-Specific Patch Embedding* module, which replaces the shared projection in standard ViTs with modality-aware embeddings adapted to multispectral and SAR data. This enables flexible handling of sensor-specific spectral profiles while preserving spatial structure. To enhance scale-invariance and cross-view consistency, we adopt multi-crop learning within a self-supervised teacher-student framework (Caron et al., 2021), promoting robust representation learning through global-local alignment. Further, we interpret different aligned modalities (S1-SAR, S2-L1C, S2-L2A) as complementary views of the same scene and introduce a *Cross-Attention Fusion* module that dynamically aggregates modality-specific tokens using learnable spatial queries. This allows the model to selectively emphasize sensor contributions at each spatial location. While modality-specific embeddings and co-located modalities have appeared in prior work (Fuller et al., 2023; Tseng et al., 2025; Bachmann et al., 2022), our

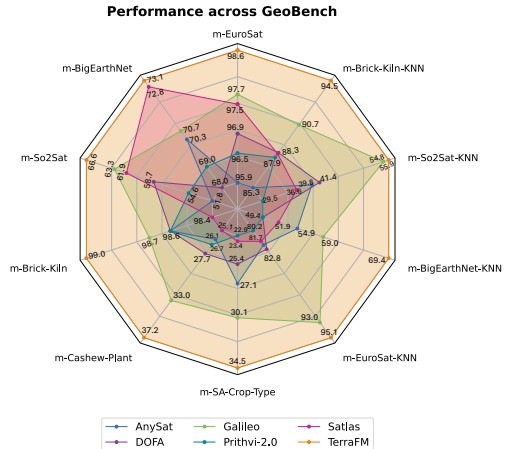

Figure 1: Performance comparison across GEO-Bench classification tasks using supervised fine-tuning and kNN evaluation. Five recent EO foundation models: AnySat (Astruc et al., 2024), DOFA (Xiong et al., 2024), Galileo (Tseng et al., 2025), Prithvi-EO-v2.0 (Szwarcman et al., 2024), and Satlas (Bastani et al., 2023) are compared against our TerraFM, which consistently outperforms them across modalities and evaluation settings, demonstrating strong generalization.

contribution lies in unifying them within a single DINO-style multi-crop backbone that treats S1, S2-L1C, and S2-L2A as complementary co-registered views, enabling stronger cross-modal coupling and richer alignment than separate-encoder designs. Moreover, prior multimodal MAE models use cross-attention for decoder-side reconstruction (Guo et al., 2024b), TerraFM performs fusion on the encoder side and treats the fused output as an additional augmented view within a single shared backbone. Finally, to mitigate long-tailed land cover distribution issues prevalent in EO data, we introduce a *Dual Centering* mechanism into the distillation process. This leverages WorldCover (Zanaga et al., 2022) derived class statistics to compute a frequency-aware center, improving balance across dominant and rare semantic categories without requiring supervised objectives. These challenges highlight the need for a unified multimodal framework that scales effectively across sensors and

Table 1: Comparison of recent remote sensing foundation models across modalities, scale (number of pretraining samples), and benchmarks. **TerraFM** uniquely blends large tile size, WorldCover-informed metadata, and global-scale training (18.7M samples) with evaluation on both GEO-Bench and Copernicus-Bench.

| Model | Modalities | Scale | Resolution | TileSize | Metadata | Benchmarks | Pixels (~T) |
|---|---|---|---|---|---|---|---|
| SatMAE++ | S2, RGB | ~1.2 M | 10–60 m | 224, 96 | No | 6 DS | 0.12 |
| Galileo | S1, S2, NDVI, ESA WC etc | ~3–10.9 M | 10 m | 96 (flex) | Yes | GEO + 5 DS | 1.58 |
| CROMA | S1, S2 | ~1 M | 10 m | 96, 120 | No | 7 DS | 0.98 |
| SoftCon | S1, S2 | ~0.78 M | 10 m | 224 | Yes | 4 GEO + 7 DS | 0.76 |
| AnySat | Aerial, S1/S2, MODIS, etc. | 11.1 M | 0.2–250 m | 10–240 | No | 11 DS | 0.17 |
| Prithvi-EO-v2.0 | S2, HLS | 4.2 M | 30 m | 224 | Yes | GEO + 9 SME | 5.06 |
| DOFA | S1, S2, EnMAP, etc. | ~8 M | 1–30 m | 512, 128 | Yes | GEO + 2 DS | 6.74 |
| Panopticon | S1, S2, WV2/3, NAIP | ~2.6 M | 0.3–100 m | 96, 224 | Yes | GEO + 10 DS | 2.34 |
| MMEarth | S1, S2, DEM, etc. | ~7.2 M | 0.3–100 m | 128 | Yes | 5 GEO | 0.51 |
| msGFM | RGB, S2, SAR, DSM | ~2 M | 0.1–30 m | 192 | No | 5 DS | 0.44 |
| Copernicus-FM | S1-S5P, DEM | 18.7 M | 10 m–1 km | Mixed | Yes | Cop-Bench | 5.12 |
| **TerraFM** (Ours) | S1, S2 L1C/L2A | 18.7 M | 10–60 m | 534 | Yes | GEO + Cop-Bench | **23.32** |

resolutions. Our approach, TerraFM, directly addresses these issues as shown in Figure 1, achieves superior performance compared to recent EO foundation models. Our key contributions are:

**Contributions: (1)** A *modality-specific patch embedding* mechanism is introduced to generalize ViTs across heterogeneous remote sensing modalities with varying spectral dimensions. **(2)** We treat sensor modalities as natural augmentations and introduce a *cross-attention fusion* block that unifies multi-modal inputs within a shared encoder. **(3)** To address long-tailed LULC distributions, a *dual-centering* strategy is incorporated to regularize representation learning using class-frequency-aware statistics. **(4)** Extensive experiments on *GEO-Bench* and *Copernicus-Bench* demonstrate leading performance across multiple downstream tasks using globally distributed data.

## 2 RELATED WORK

**Self-supervised Pretraining:** MAEs (He et al., 2022) have become a popular choice for self-supervised pretraining in remote sensing by reconstructing masked image regions using ViT (Dosovitskiy et al., 2021). Variants like Scale-MAE (Reed et al., 2023) and MC-MAE (Gao et al., 2022) enhance robustness across spatial scales via scale-aware encodings and convolutional tokenizers. However, MAEs struggle to scale to multisensor EO data, as their RGB-centric tokenization and reconstruction objectives limit generalization to multispectral and SAR modalities with diverse channel structures (Xie et al., 2023; Li et al., 2024). Unlike MAEs, self-supervised contrastive learning focuses on learning discriminative representations by comparing semantically similar and dissimilar views.

Remote sensing approaches Tang et al. (2023); Fuller et al. (2023); Waldmann et al. (2025) leverage spatial and spectral augmentations to create diverse yet consistent views. CROMA (Fuller et al., 2023) combines contrastive and masked autoencoding losses, while Cross-Scale MAE (Tang et al., 2023) blends generative and contrastive objectives for multi-scale learning. Student-teacher frameworks like DINO (Caron et al., 2021; Oquab et al., 2023) scale contrastive learning via EMA-updated teachers and global-local view alignment with centering to prevent collapse. These strategies are well-suited for EO, where multimodal imagery can act as natural augmentations, enabling scalable, label-free training and broad generalization.

**Remote Sensing FMs:** Recent advances in remote sensing foundation models (FMs) have scaled self-supervised learning across architecture types, modalities, training sizes, tile resolutions, and metadata usage (Table 1). Multimodal integration is central to recent FMs like Guo et al. (2024a); Wang et al. (2025); Waldmann et al. (2025); Astruc et al. (2024); Tseng et al. (2025); Han et al. (2024). SkySense (Guo et al., 2024a) applies contrastive learning to temporal-multimodal data but requires large-scale compute. CopernicusFM (Wang et al., 2025) fuses Sentinel modalities via metadata-aware networks but faces scaling issues with heterogeneous inputs. Panopticon (Waldmann et al., 2025) and AnySat (Astruc et al., 2024) align cross-modal views through contrastive training, while Galileo (Tseng et al., 2025) uses shared embeddings for SAR and multispectral fusion. Fus-MAE (Chan-To-Hing & Veeravalli, 2024) adopts attention-based fusion without contrastive loss, limiting generalization.

Prithvi-EO-v2.0 (Szwarcman et al., 2024) is restricted to single-modal optical data with temporal-spatial modeling. DOFA (Xiong et al., 2024), msGFM (Han et al., 2024), and AnySat (Astruc et al.,

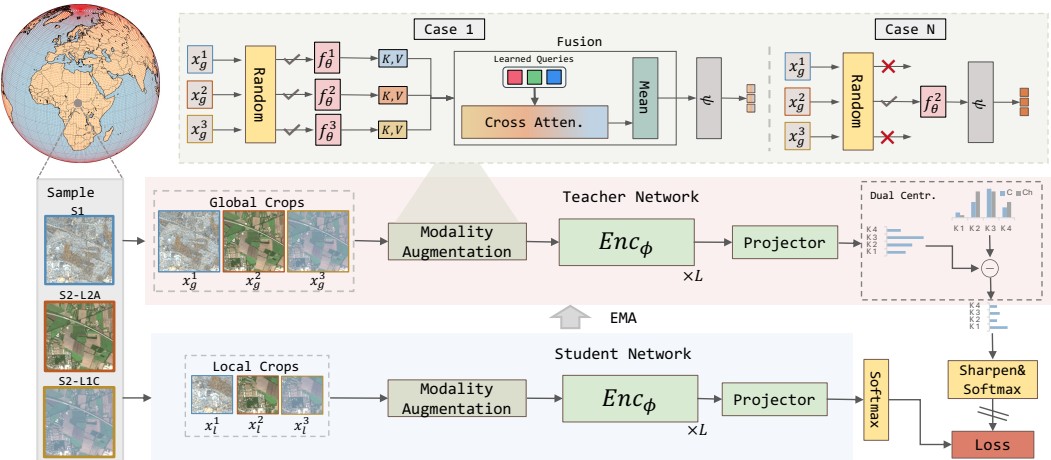

Figure 2: Overall architecture of **TerraFM**. It unifies student-teacher contrastive framework with modality augmentation with cross-attention fusion, and a new dual centering regularization. **TerraFM** is founded on ViT backbone and is trained on 18.7M globally distributed samples for pre-training and utilizes large-tile inputs for encoding broader spatial context. For *illustration*, RGB channels from S2-L2A and S2-L1C are selected, and S1 is visualized using a false-color RGB composite.

2024) address resolution variability using mixed tile sizes or scale-adaptive designs. Our 534px tiles capture broader spatial context than prior RSFMs. While CopernicusFM (Wang et al., 2025) and DOFA (Xiong et al., 2024) incorporate metadata, we leverage land cover (LULC) priors for semantically informed learning. Both CopernicusFM and our model are trained on 18.7M samples, but ours uses over 23T pixels during pretraining, scaling the 5.1T used by Copernicus-Pretrain (Wang et al., 2025) over 4x. This growing body of work makes clear that the next step is moving beyond single-modality or scale-limited pipelines toward unified, globally robust EO foundation models.

## 3 TERRAFM: A SCALABLE MULTISENSOR FOUNDATIONAL MODEL

Unlike prior remote sensing foundation models, our approach integrates a student–teacher contrastive learning framework with dual centering (to balance long-tailed classes), modality-as-augmentation (to learn cross-modal invariances), and cross-attention fusion (to aggregate multi-sensor context), as illustrated in Figure 2. Built on a ViT backbone and trained on 18.7M globally distributed samples using 534×534 tiles, **TerraFM** captures broader spatial context and generalizes effectively across sensing modalities and geographies, achieving strong results on diverse downstream benchmarks.

### 3.1 ARCHITECTURE

We use globally distributed remote sensing imagery organized over a spatial grid, partitioning the earth's surface into fixed-size tiles (Francis & Czerkawski, 2024). Each spatial unit, denoted as $s$, represents one such grid cell. For each sample, we observe a set of co-registered EO modalities:

$$\mathcal{M} = \{\text{S1, S2-L1C, S2-L2A}\},$$

where **S1** corresponds to Sentinel-1 SAR, and **S2-L1C** and **S2-L2A** represent two processing levels of Sentinel-2 optical imagery: Level-1C (top-of-atmosphere reflectance) and Level-2A (bottom-of-atmosphere surface reflectance), respectively. Each modality $m \in \mathcal{M}$ provides a multi-channel image $\boldsymbol{x}^m \in \mathbb{R}^{H \times W \times C_m}$, where $H$ and $W$ denote spatial dimensions, and $C_m$ is the number of spectral channels for modality $m$. For example, Sentinel-1 contains two channels (VV and VH polarizations), therefore $C_{\text{S1}} = 2$, while Sentinel-2 modalities contain up to 13 spectral bands depending on level and resolution. These modalities are treated as complementary views of the same location, acting as natural augmentations, which support our training strategy and encourage learning modality-invariant representations. To provide semantic grounding, each sample $s$ is assigned a high-level land use and land cover (LULC) category $y^{(s)} \in \{1, \ldots, Y\}$, derived from the ESA WorldCover product. These categories reflect coarse semantic classes at a global scale and are used to compute class-frequency-aware statistics for balanced representation learning.

**Vision Transformer Model:** ViTs adapt the transformer architecture to visual data by treating an image as a sequence of patch tokens instead of a dense pixel grid. A typical ViT consists of two main components: a patch embedding module and a transformer encoder. Given an input image $\boldsymbol{x} \in \mathbb{R}^{H \times W \times C}$, the **patch embedding** layer $f_\theta$ divides the image into $N$ non-overlapping patches of size $P \times P$, and projects each patch into a $d$ dimensional embedding:

$$\{\boldsymbol{z}_i\}_{i=1}^N = f_\theta(\boldsymbol{x}), \qquad \boldsymbol{z}_i \in \mathbb{R}^d.$$

This projection is typically implemented using a convolutional layer with kernel size and stride equal to the patch size $P$, parameterized by weights $\mathbf{W}_\theta \in \mathbb{R}^{d \times C \times P \times P}$. To encode spatial information, the transformer encoder augments each patch token $\boldsymbol{z}_i$ with a positional vector. A learnable class token $\boldsymbol{z}_{\mathrm{cls}}$ is added to the sequence, which yields the full input:

$$\boldsymbol{Z} = \big[\, \boldsymbol{z}_{\mathrm{cls}};\ \{\boldsymbol{z}_i + \mathrm{pos}_i\}_{i=1}^N \,\big].$$

The token sequence $\boldsymbol{Z}$ is processed by a stack of $L$ transformer layers, denoted $\mathrm{Enc}_\phi$. For classification tasks, only the final class token $\boldsymbol{z}_{\mathrm{cls}}$ is forwarded to a prediction head.

**Modality-Specific Patch Embedding:** Standard patch embedding layers in ViTs are typically implemented using a shared convolutional projection across all inputs, making it unsuitable for multi-modal remote sensing data. To better handle this heterogeneity, we adopt a modality-specific patch embedding strategy. For each modality $m \in \mathcal{M}$, we define an embedding function $f_{\theta_m}$ that maps the input image $\boldsymbol{x}^{(m)} \in \mathbb{R}^{H \times W \times C_m}$ to a sequence of patch tokens $\bar{\boldsymbol{Z}}^{(m)} \in \mathbb{R}^{N \times D}$, where $C_m$ is the number of channels and $N$ is the number of patches. Each $f_{\theta_m}$ is parameterized independently to account for modality-specific dynamics. We associate each modality with a learnable embedding vector $\boldsymbol{\epsilon}^{(m)} \in \mathbb{R}^D$. This vector is added to every token from that modality via broadcasting:

$$\tilde{\boldsymbol{Z}}^{(m)} = \bar{\boldsymbol{Z}}^{(m)} + \mathbf{1}_N \cdot (\boldsymbol{\epsilon}^{(m)})^\top,$$

where $\mathbf{1}_N \in \mathbb{R}^{N \times 1}$ is a vector of ones. This allows the model to distinguish between modalities while preserving local spatial and spectral features. Finally, to enable shared processing in the Transformer encoder, the enriched tokens $\tilde{\boldsymbol{Z}}^{(m)}$ are linearly projected into a common latent space of dimension $d$ using a shared projection $\psi : \mathbb{R}^D \to \mathbb{R}^d$:

$$\boldsymbol{Z}^{(m)} = \psi(\tilde{\boldsymbol{Z}}^{(m)}) \in \mathbb{R}^{N \times d}.$$

This operation aligns all modality-specific token sequences in a unified representation space, allowing the encoder to process them jointly.

**Modality Augmentation and Cross-Attention Fusion:** Remote sensing observations of a single location are often captured using multiple sensors, each providing a unique spectral or radiometric perspective. Instead of treating these modalities as independent inputs, we interpret them as complementary views of the same scene. This allows us to use modality diversity as a form of natural augmentation, enabling the model to learn sensor-invariant representations. In our setup, each spatial sample $s$ from the Major-TOM dataset (Francis & Czerkawski, 2024) is observed via a fixed set of modalities. During pretraining, we independently assign modalities to the student and teacher networks via stochastic selection (threshold = 0.5), ensuring cross-modal supervision. E.g., the teacher may observe a global crop from Sentinel-1, while the student receives local views from Sentinel-2 L2A. This modality augmentation strategy encourages the model to align features across sensors, improving robustness to sensor-specific artifacts. We consider two cases based on the number of selected modalities:

**1) Single-Modality Views:** If only one modality is selected, the input is passed through the corresponding modality-specific patch embedding layer followed by the shared transformer encoder. This follows the standard ViT pipeline but uses modality-aware embeddings to handle spectral channel differences. **2) Multi-Modality Fusion via Cross-Attention:** When multiple modalities $M \subseteq \mathcal{M}$ are selected, we activate a modality fusion module based on cross-attention. For each selected modality $m \in M$, we obtain a patch token sequence $\bar{\boldsymbol{Z}}^{(m)} \in \mathbb{R}^{N \times D}$, where $N$ is the number of spatial positions. These are stacked into a tensor $\boldsymbol{Z}_{\mathrm{all}} \in \mathbb{R}^{N \times M \times D}$, aligning spatial positions across modalities.

For each position $n = 1, \ldots, N$, we define shared learnable queries $\boldsymbol{q} \in \mathbb{R}^{N_q \times D}$, which attend to modality-specific keys $\boldsymbol{K}_n \in \mathbb{R}^{M \times D}$ and values $\boldsymbol{V}_n \in \mathbb{R}^{M \times D}$, where, $\boldsymbol{K}_n$, and $\boldsymbol{V}_n$ are obtained via separate learned linear projections of modality-specific tokens, yielding $N_q$ intermediate outputs:

$$\boldsymbol{z}'_n = \mathtt{MultiHeadAttention}(\boldsymbol{q}, \boldsymbol{K}_n, \boldsymbol{V}_n) \in \mathbb{R}^{N_q \times D}.$$

To aggregate them, we compute a learned weighted mean using softmax-normalized attention scores:

$$\boldsymbol{w} = \texttt{Softmax}(\boldsymbol{z}_n' \cdot \boldsymbol{p}_r), \quad \boldsymbol{z}_n^{\text{fused}} = \sum_{j=1}^{N_q} w_j \boldsymbol{z}_n'[j],$$

where $\boldsymbol{p}_r \in \mathbb{R}^{D \times 1}$ is a learnable projection for scoring the query outputs. This results in a fused token $\boldsymbol{z}_n^{\text{fused}} \in \mathbb{R}^D$. The final sequence $\boldsymbol{Z}_{\text{fused}} \in \mathbb{R}^{N \times D}$ is then passed to the shared encoder $\text{Enc}_\phi$. This cross-attention fusion allows the model to dynamically weigh the modality contributions at each spatial location, capturing diverse information while maintaining spatial coherence. For clarity, the learnable queries are shared across all spatial locations, and the fusion yields one token per location, preserving the ViT backbone's original sequence length.

## 3.2 PRETRAINING

Our pretraining strategy builds on the DINO framework, which performs self-supervised learning. It operates using a teacher-student setup, where both networks share the same ViT backbone and a lightweight three-layer projection head. Let $g_{\theta_s}$ and $g_{\theta_t}$ denote the student and teacher networks, respectively. While the student is trained using gradient-based optimization, the teacher is updated using EMA of the student's weights:

$$\theta_t \ \leftarrow \ \lambda_e \, \theta_t + (1 - \lambda_e) \, \theta_s, \quad \lambda_e = 1 - (1 - \lambda_0) \frac{1 + \cos(\pi e / E)}{2}, \tag{1}$$

where $e$ is the current epoch, $E$ is the total number of training epochs, and $\lambda_0 \in [0.996, 1)$ is the initial momentum coefficient. The cosine schedule gradually increases $\lambda_e$, stabilizing the teacher updates as training progresses. This EMA mechanism allows the teacher to serve as a temporally smoothed ensemble of past student states, yielding more stable and consistent targets.

**Multi-Crop Learning:** To enable scale-invariant and cross-view representation learning, we adopt a multi-crop strategy as used in DINO (Caron et al., 2021). For each input sample, we generate two high-resolution global crops $\{\boldsymbol{x}_g^{(1)}, \boldsymbol{x}_g^{(2)}\} \subset \mathcal{X}_g$ and $J$ low-resolution local crops $\{\boldsymbol{x}_\ell^{(j)}\}_{j=1}^J \subset \mathcal{X}_\ell$. The teacher network processes only the global crops, while the student receives both global and local views. Each network produces a $K$-dim output which is temperature-scaled and normalized via the softmax function:

$$Q_s^{(i)}(\boldsymbol{x}) = \frac{\exp(g_{\theta_s}(\boldsymbol{x})^{(i)}/\tau_s)}{\sum_{k=1}^K \exp(g_{\theta_s}(\boldsymbol{x})^{(k)}/\tau_s)}, \quad Q_t^{(i)}(\boldsymbol{x}) = \frac{\exp((g_{\theta_t}(\boldsymbol{x})^{(i)} - c^{(i)})/\tau_t)}{\sum_{k=1}^K \exp((g_{\theta_t}(\boldsymbol{x})^{(k)} - c^{(k)})/\tau_t)},$$

where $\tau_s$ and $\tau_t$ are temperature parameters that control output sharpness, and $c \in \mathbb{R}^K$ is a centering term representing the running mean of teacher logits, used to stabilize training and avoid representation collapse. The centering term is updated using an exponential moving average over the teacher outputs:

$$c \leftarrow \beta c + (1 - \beta) \cdot \frac{1}{B} \sum_{i=1}^B g_{\theta_t}(\boldsymbol{x}_i),$$

where $\beta \in [0.9, 0.999]$ controls the momentum, and $B$ is the batch size. The overall loss encourages consistency between teacher and student predictions across all distinct view pairs:

$$\sum_{\boldsymbol{x} \in \mathcal{X}_g} \sum_{\boldsymbol{x}' \in \mathcal{X}; \boldsymbol{x}' \neq \boldsymbol{x}} \mathcal{L}_{\text{CE}}\left(Q_t(\boldsymbol{x}), Q_s(\boldsymbol{x}')\right),$$

where $\mathcal{X} = \mathcal{X}_g \cup \mathcal{X}_\ell$, and $\mathcal{L}_{\text{CE}}(\cdot, \cdot)$ denotes the cross-entropy loss. This loss formulation requires the student to produce consistent representations in all views.

**Dual Centering for Long-Tailed Distributions:**

Remote sensing datasets often exhibit long-tailed distributions of LULC classes, with frequent categories such as Forest dominating, while classes like Urban or Bare Land remain underrepresented as shown in Figure 3. This imbalance persists even after subsampling and poses challenges for representation learning. Standard self-supervised approaches like DINO (Caron et al., 2021) apply a single global centering term to stabilize training and avoid representation collapse, but they do not

account for semantic imbalance in the data. To address this, we propose a dual-centering scheme that combines global statistics with class-frequency-aware regularization. In addition to the standard global center vector $c$, we introduce a secondary center $c_h \in \mathbb{R}^K$, computed from a subset of samples belonging to high-frequency LULC classes, such as tree cover, grassland, and open seas, based on dataset-level statistics. Given a batch of teacher logits $g_{\theta_t}(\boldsymbol{x})$, the adjusted logits for training are computed as:

$$\hat{g}(\boldsymbol{x}) = g_{\theta_t}(\boldsymbol{x}) - \alpha \cdot c - (1 - \alpha) \cdot c_h,$$

where $\alpha \in [0, 1]$ balances the contribution of the global and frequency-aware centers. The vector $c_h$ is updated via exponential moving average using only frequent-class samples within each batch. This dual-centering mechanism serves two key purposes: **(i)** it preserves the stability benefits of global centering as in DINO, and **(ii)** it introduces a soft rebalancing bias that counteracts the overrepresentation of dominant classes in the feature space. In ablations (Table 5), this adjustment leads to more balanced representation learning and improved downstream performance, particularly for underrepresented LULC categories.

## 4    PRETRAINING DATA SAMPLING

We utilize Major-TOM dataset (Francis & Czerkawski, 2024) as our primary EO source for pertaining. It contains 2.24 million globally distributed grid cells, each spanning approximately 10.68 km × 10.68 km (≈114 km²), and provides tri-modal, co-registered imagery from Sentinel-2 Level-1C, Sentinel-2 Level-2A, and Sentinel-1 RTC. Major-TOM stands out as one of the few publicly available datasets offering dense multi-modal coverage at a global scale. Each cell independently selects a random 4-month window before cloud screening, which helps limit systematic seasonal and regional bias. However, over one-third of its samples lie outside a 10 km terrestrial buffer, often within the Open Oceans class (Zanaga et al., 2022), limiting their relevance for land-centric tasks. Motivated by insights from Roscher et al. (2024), which emphasize the importance of semantically rich samples, and Wang et al. (2024d), which highlight the utility of structural priors, we applied a principled filtering strategy. Specifically, we removed 98% of ocean-classified tiles (retaining 2% to preserve marine representation) and sampled the terrestrial subset using global distributional priors across land cover (Zanaga et al., 2022), climate zones (Beck et al., 2018), and ESRI world regions (Esri et al., 2025). Global distributional priors refer specifically to the majority land-cover label for each grid cell, which we retain only to compute class-frequency statistics for the dual-centering regularizer; this metadata is not provided as model input or used as supervision. This approach emphasizes meaningful land regions with ecological variety. Figure 3 shows the global coverage of the tile and Figure A4 in the Appendix covers the map. For pretraining, we curated a filtered subset of over 1.5 million grid cells with consistent coverage across all three modalities (S1, S2-L1C, and S2-L2A). Each 10.68 km × 10.68 km grid cell was divided into four non-overlapping tiles of 534 × 534 pixels, resulting in more than 6 million tiles per modality. In total, this yielded 18.7 million modality-specific training tiles. During training, modalities were stochastically sampled and treated as natural augmentations to promote sensor-invariant representation learning. To mitigate spatial sampling bias and support semantically-aware learning, we enriched each grid cell with metadata from the ESRI World Regions dataset (Esri et al., 2025).

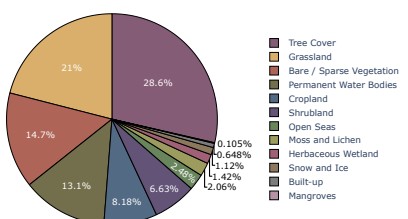

Figure 3: Land-use/land-cover (LULC) breakdown across the training tiles. A number of semantically important classes (e.g., builtup, mangroves, ice) remain underrepresented due to skewed data distribution.

## 5    EXPERIMENTS AND RESULTS

### 5.1    PRETRAINING IMPLEMENTATION DETAILS

We pre-train TerraFM using 534 × 534 tiles as inputs. Following the DINO-style cropping strategy (Caron et al., 2021), each tile is randomly cropped at two scales: (i) *global crops*, sampled with ratios in $[0.25, 1.0]$ of the tile size and resized to 224 × 224, and (ii) *local crops*, sampled with

Table 2: We evaluate image classification using k-nearest neighbors (kNN) and report Top-1 accuracy for all single-label tasks. For the multilabel BigEarthNet benchmark, we report the F1 score. Results other than Copernicus-FM and TerraFM are directly taken from Tseng et al. (2025). **Bold** indicates the best result, and underlining denotes the second-best.

| Model | Backbone | m-EuroSat Training % 100% | 1% | m-BigEarthNet Training % 100% | 1% | m-So2Sat Training % 100% | 1% | m-Brick-Kiln Training % 100% | 1% |
|---|---|---|---|---|---|---|---|---|---|
| SatMAE (Cong et al., 2022) | ViT-Base | 84.1 | 34.8 | 50.6 | 29.0 | 36.0 | 23.1 | 86.1 | 73.5 |
| SatMAE++ (Noman et al., 2024) | ViT-Large | 82.7 | 48.5 | 50.8 | 31.6 | 34.7 | 23.4 | 89.6 | 76.7 |
| CROMA (Fuller et al., 2023) | ViT-Base | 85.6 | 51.3 | 58.8 | 44.7 | 48.8 | 33.8 | 92.6 | 85.1 |
| SoftCon (Wang et al., 2024b) | ViT-Small | 89.8 | 27.2 | 64.7 | 43.3 | 51.1 | 31.4 | 89.2 | 77.8 |
| DOFA (Xiong et al., 2024) | ViT-Base | 82.8 | 49.6 | 49.4 | 29.9 | 41.4 | 29.4 | 88.3 | 78.3 |
| Satlas (Bastani et al., 2023) | Swin-Tiny | 81.7 | 35.8 | 51.9 | 29.6 | 36.6 | 27.1 | 88.2 | 73.0 |
| MMEarth (Nedungadi et al., 2024) | CNN-atto | 81.7 | 30.0 | 58.3 | 39.6 | 39.8 | 25.1 | 89.4 | 79.7 |
| DeCUR (Wang et al., 2024a) | ViT-Small | 89.0 | 46.6 | 63.8 | 49.6 | 45.8 | 30.9 | 83.7 | 74.2 |
| AnySat (Astruc et al., 2024) | ViT-Base | 82.2 | 47.1 | 54.9 | 33.7 | 39.8 | 29.0 | 85.3 | 72.0 |
| Galileo (Tseng et al., 2025) | ViT-Base | 93.0 | 56.6 | 59.0 | 36.5 | 54.8 | 43.2 | 90.7 | 78.0 |
| Prithvi-EO-v2.0 (Szwarcman et al., 2024) | ViT-Large | 80.2 | 48.0 | 49.4 | 28.8 | 29.5 | 26.1 | 87.9 | 80.6 |
| Copernicus-FM(Wang et al., 2025) | ViT-Base | 76.0 | 47.4 | 53.8 | 33.3 | 38.4 | 23.3 | 93.0 | 83.2 |
| TerraFM (Ours) | ViT-Base | 94.2 | 59.3 | 68.7 | 49.4 | 55.1 | 41.6 | 94.5 | 85.6 |
| | ViT-Large | 95.1 | 62.1 | 69.4 | 50.6 | 55.9 | 41.1 | 93.0 | 82.2 |

Table 3: Performance comparison on GEO-Bench for both classification (Top-1 Accuracy), segmentation (mIoU), and F1 score (for m-BigEarthNet). TerraFM achieves state-of-the-art results across multiple datasets, outperforming previous FMs. **Bold** indicates the best result, and underlining denotes the second-best.

| Method | Backbone | Classification m-EuroSat | m-BigEarthNet | m-So2Sat | m-Brick-Kiln | Segmentation m-Cashew-Plant | m-SA-Crop-Type |
|---|---|---|---|---|---|---|---|
| SatMAE | ViT-Large | 96.6 | 68.3 | 57.2 | 98.4 | 30.8 | 24.8 |
| SatMAE++ | ViT-Large | 96.5 | 67.9 | 56.0 | 98.6 | 29.6 | 25.7 |
| CROMA | ViT-Large | 96.6 | 71.9 | 60.6 | 98.7 | 31.8 | 32.0 |
| SoftCon | ViT-Base | 97.5 | 70.3 | 61.7 | 98.7 | 29.6 | 30.8 |
| DOFA | ViT-Large | 96.9 | 68.0 | 58.7 | 98.6 | 27.7 | 25.4 |
| Satlas | Swin-Base | 97.5 | 72.8 | 61.9 | 98.4 | 25.1 | 23.4 |
| MMEarth | CNN-atto | 95.7 | 70.0 | 57.2 | 98.9 | 24.2 | 22.2 |
| DeCUR | ViT-Small | 97.9 | 70.9 | 61.7 | 98.7 | 26.2 | 21.5 |
| Prithvi-EO-v2.0 | ViT-Large | 96.5 | 69.0 | 54.6 | 98.6 | 26.7 | 22.9 |
| AnySat | ViT-Base | 95.9 | 70.3 | 51.8 | 98.6 | 26.1 | 27.1 |
| Galileo | ViT-Base | 97.7 | 70.7 | 63.3 | 98.7 | 33.0 | 30.1 |
| TerraFM | ViT-Base | 98.1 | 72.6 | 64.9 | 98.7 | 34.1 | 32.8 |
| | ViT-Large | 98.6 | 73.1 | 66.6 | 99.0 | 37.0 | 34.6 |

ratios in $[0.05, 0.25]$ and resized to $96 \times 96$. All inputs are then tokenized with a $16 \times 16$ patch resolution. The training dataset comprises around 1.53 million multi-modal samples, from which we define a virtual epoch of 300K samples to ensure frequent parameter updates. TerraFM-B is trained for 150 epochs and TerraFM-L for 200 epochs, each with a linear warmup over the first 30 epochs. Models are trained on 64 GPUs, TerraFM-B training takes 92 hours with a batch size of 1024, while TerraFM-L uses a batch size of 2048 and trains for 183 hours. The learning rate is linearly scaled with batch size, initialized as lr $= 0.0001 \times$ batch_size$/256$. Following DINO-style pretraining, we disable batch normalization in the projection head and freeze the last layer of the student for the first 3 epochs to stabilize early training. The output dimensionality is set to $K = 65{,}536$, with the teacher temperature linearly increasing from 0.04 to 0.06 over the first 50 epochs. The teacher momentum follows a cosine schedule starting from 0.996. A drop path rate of 0.1 is applied for regularization. For modality fusion, we set $N_q = 5$ and $\alpha = 0.8$ during pretraining.

## 5.2 EVALUATING DOWNSTREAM TASKS

**Benchmarks:** We evaluate our model on two comprehensive remote sensing benchmarks: **GEO-Bench** (Lacoste et al., 2023) and **Copernicus-Bench** (Wang et al., 2024a), both of which include diverse downstream tasks spanning multiple domains and modalities. The benchmark datasets are described in Appendix A, and for evaluation protocols (linear probing, UperNet probing, k-NN, and fine-tuning), we refer the reader to Appendix C.1.

**Discussion:** We report KNN classification accuracy on four standard GEO-Bench classification tasks to evaluate the quality of learned representations in a training-free setting. As shown in Table 2, TerraFM achieves the highest performance across three datasets, outperforming both modality-specific and multimodal foundation models. Notably, our model achieves 95.1% on m-EuroSAT and

94.5% on m-Brick-Kiln, highlighting the effectiveness of the learned representations on standard scene classification tasks. On other challenging tasks such as m-So2Sat and m-BigEarthNet, our model achieves leading performance (55.9% and 69.4%, respectively), outperforming Galileo (Tseng et al., 2025), despite So2Sat having fewer channels than used during pretraining, highlighting the model's robustness to missing modality information. Compared to CROMA (Fuller et al., 2023) and DeCUR (Wang et al., 2024a), our gains suggest that contrastive alignment combined with cross-modal fusion enhances class separability. The results across tasks of varying difficulty indicate that our model learns robust and transferable representations that generalize well across different scenarios.

Further GEO-Bench results with fine-tuning and linear probing are reported in Table 3, for classification (with fine-tuning), TerraFM achieves the improvement on m-BigEarthNet (73.1%) and m-EuroSat (98.6%), and the best-performing model on m-So2Sat (66.6%). For segmentation (with linear probing), our TerraFM-L notably outperforms existing models on m-SA-Crop-Type (34.6% mIoU) and m-Cashew-Plant (37.0% mIoU). TerraFM-B surpasses larger counterparts such as ViT-Large used in SatMAE++(Noman et al., 2024) and DOFA(Xiong et al., 2024). On the Copernicus-Bench (Wang et al., 2025) (Table 4), TerraFM delivers state-of-the-art results across most tasks and modalities. It achieves 67.9 mIoU on Cloud-S2, 87.8 OA on EuroSAT-S1, and 99.1 OA on EuroSAT-S2, surpassing all prior models. On BigEarthNet-S2 and DFC2020-S1, TerraFM attains 84.4 mAP and 55.4 mIoU, respectively, marking clear gains over existing FMs. While SoftCon (Wang et al., 2024b) is slightly higher on BigEarthNet-S1 (78.7 vs. 76.9), TerraFM consistently outperforms Copernicus-FM (Wang et al., 2025) despite sharing the same ViT-B/16 backbone. These results highlight TerraFM's scalability and strong generalization across diverse EO benchmarks.

Table 4: Comparison of TerraFM with supervised and self-supervised methods on Copernicus-Bench. Metrics: OA (Overall Accuracy for classification), mAP (mean Average Precision for multi-label classification), mIoU (mean Intersection over Union for segmentation). Baselines include SoftCon (Wang et al., 2024b), CROMA (Fuller et al., 2023), DOFA (Xiong et al., 2024), and Copernicus-FM (Wang et al., 2025). **Bold** indicates the best result, and underlining denotes the second-best.

| Method | Backbone | Classification | | | | | Segmentation | | |
|---|---|---|---|---|---|---|---|---|---|
| | | EuroSAT-S1 | EuroSAT-S2 | BigEarthNet-S1 | BigEarthNet-S2 | LCZ-S2 | Cloud-S2 | DFC2020-S1 | DFC2020-S2 |
| Supervised | ViT-B/16 | 81.5 | 97.6 | 70.6 | 80.1 | 85.3 | 59.4 | 50.8 | 66.2 |
| Random | ViT-B/16 | 75.4 | 92.5 | 63.8 | 71.6 | 77.4 | 60.4 | 45.4 | 62.3 |
| SoftCon | ViT-B/14 | 83.6 | 96.7 | **78.7** | 83.6 | 83.6 | 66.9 | 52.8 | 64.1 |
| CROMA | ViT-B/8 | 83.9 | 97.0 | 70.8 | 76.4 | 84.1 | 65.0 | 52.7 | **66.5** |
| DOFA | ViT-B/16 | 81.7 | 97.2 | 70.5 | 75.5 | 83.0 | 65.0 | 49.7 | 61.8 |
| Copernicus-FM | ViT-B/16 | 87.2 | 97.9 | 77.9 | 79.0 | 84.4 | 66.7 | 52.4 | 64.5 |
| TerraFM | ViT-B/16 | **87.8** | **99.1** | 76.9 | **84.4** | **87.0** | **67.9** | **55.4** | 63.8 |

## 5.3 ABLATIONS AND ANALYSIS

**Impact of Components:** Table 5 highlights the incremental benefits of each component in our framework. We train TerraFM-B for 150 epochs on a 200k-sample subset from our full training dataset. To measure the performance on segmentation task, we use uppernet probing and linear probing on the m-Cashew-Plantation dataset from GeoBench. Adding modality as augmentation improves performance on m-EuroSat by +4.5% and m-BigEarthNet by +3.01%. Incorporating fusion yields a large gain on m-Cashew-Plantation segmentation by +3.23% with UPerNet probing and +1.4% with linear probing, while dual centering provides further improvements: +0.32% on m-BigEarthNet, +1.9% on m-EuroSat, and +2.18% on m-Cashew-Plantation. Note that in Table 5, the dual-centering ablation disables only our additional centering term; the standard DINO global centering remains active in all configurations to maintain the stability of the student–teacher training dynamics.

**Dual-centering Motivation and Visualization:** Here, we discuss the impact of Dual-centering on class-wise prediction behavior and representation diversity.

Figure 4 shows that models with Dual-centering exhibit higher softmax entropy across most classes, indicating more calibrated predictions, particularly benefiting rare classes like "Mangroves". Figure 5 reveals that Dual Centering significantly increases prototype diversity, i.e., the number of distinct top-5 features activated, especially for tail classes. This suggests that the model avoids collapsing onto frequent-class prototypes and learns more diverse, semantically rich representations. These results

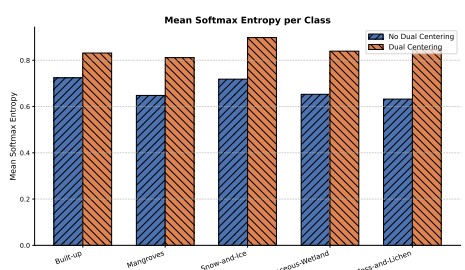 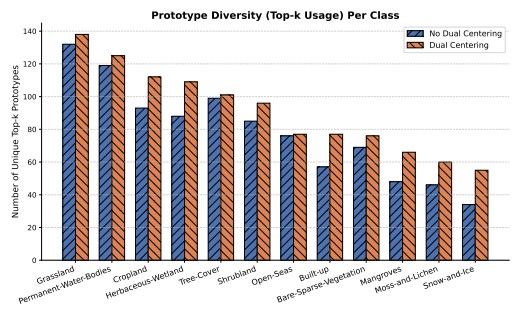

Figure 4: Mean entropy per LULC class from 5k training sam- ples, with logits ($K = 65, 536$) reduced via Gaussian projection. The baseline (no dual-centering) shows lower entropy and overconfident predictions skewed to frequent classes. Dual-centering increases entropy, yielding more balanced predictions, especially for rare classes like Mangroves and Herbaceous-Wetland.

Figure 5: Prototype diversity is measured as the number of unique top-5 prototypes across 5k training samples. Dual- centering improves diversity for tail classes like Mangroves, Herbaceous-Wetland, and Built-up, indicating richer represen- tations. The baseline reuses fewer prototypes, reflecting over- reliance on dominant frequent-class features.

| SS | MAug | Fus | DC | BEN | ES | CP† | CP‡ |
|---|---|---|---|---|---|---|---|
| ✓ | – | – | – | 54.62 | 83.20 | 50.58 | 19.4 |
| ✓ | ✓ | – | – | 57.63 (+3.01) | 87.70 (+4.50) | 59.17 (+8.59) | 24.8 (+5.4) |
| ✓ | ✓ | ✓ | – | 57.74 (+0.11) | 88.50 (+0.80) | 62.40 (+3.23) | 26.2 (+1.4) |
| ✓ | ✓ | ✓ | ✓ | 58.06 (+0.32) | 90.40 (+1.90) | 64.58 (+2.18) | 27.6 (+1.4) |

Table 5: Ablation of components: SS = Self-supervised contrastive learning, MAug = Modality Augmentation, Fus = Fusion, DC = Dual-Centering. BEN = m-BigEarthNet, ES = m-EuroSat, CP = m-Cashew-Plant. † denote results using UPerNet probing while ‡ indicate linear probing. Gains in parentheses denote improvements over previous row.

motivate Dual-centering as an effective strategy for reducing class imbalance effects in representation learning.

**MACs-Performance Trade-Off:** We analyze the compute–accuracy trade-off using Multi-ply–Accumulate operations (MACs) to measure inference cost. As shown in Figure 6, TerraFM achieves the highest m-EuroSat accuracy at substantially lower MACs, validating the efficiency of our fusion design and pretraining strategy. Notably, models with higher MACs do not guarantee better performance, underscoring the need for compact yet expressive architectures in scalable EO settings. For further results and other supporting information, we refer readers to the Appendix.

## 6 CONCLUSION

In this work, we introduced TerraFM, a unified and scalable foundation model (FM) specifically designed for multisensor EO. Given the unique nature of EO data, our approach explicitly accounts for sensor heterogeneity, scale-invariance, and class-frequency imbalance which is critical for building generalizable EO FMs. Our pretraining approach leverages contrastive learning to obtain geographically and spectrally aware representations from large-scale Sentinel-1 and 2 data. Specifically, we integrate modality-specific patch embeddings, adaptive cross-attention fusion, and a dual-centering contrastive learning objective to enrich the representations on heterogeneous RS data. Our extensive evaluations on GEO-Bench and Copernicus-Bench demonstrate that TerraFM consistently outperforms SoTA self-supervised ViT models across both classification and segmentation tasks.

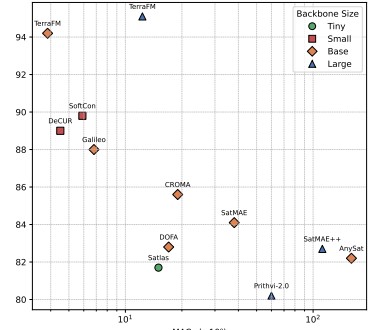

Figure 6: Comparison of model efficiency and accuracy on m-EuroSat. We plot the MACs against k-NN classification accuracy for various baselines. TerraFM models achieve the highest accuracy while maintaining moderate computational cost.

## ACKNOWLEDGMENTS

We acknowledge the LUMI supercomputer, owned by the EuroHPC Joint Undertaking and hosted by CSC, and the LUMI consortium, for providing the computational resources that enabled this work.

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

APPENDIX

This supplementary material presents additional experiments, analyses, and visualizations that complement the main paper. It includes benchmarks information (A), detailed descriptions & experiments for our multimodal fusion strategies (B), implementation details (C.1) and evaluations on scaling trends with data size (C.2), high resolution benchmarks (C.3), & change detection (C.4), Further more we also provide additional analysis and qualitative figures (D.1). We also report GPU-hour comparisons with comparable methods (D.2), landslide detection (D.3) and visualize the land cover distribution of our dataset using global maps (D.4).

## A  BENCHMARKS

We evaluate TerraFM on two comprehensive remote sensing benchmarks, **GEO-Bench** and **Copernicus-Bench**, which together cover a wide range of tasks, modalities, and resolutions. **1) GEO-Bench** (Lacoste et al., 2023) standardizes evaluation across 12 curated tasks, including 6 classification and 6 segmentation challenges. These datasets are selected for open access and license compliance, and have been harmonized with consistent evaluation settings. While GEO-Bench supports various sensor types (e.g., Landsat-8, Sentinel-2, and hyperspectral sensors), for consistency with our model's pretraining, we restrict the main evaluation to Sentinel-2-based tasks. Specifically, we report results on *m-EuroSAT (Helber et al., 2019), m-BigEarthNet (Sumbul et al., 2019), m-So2Sat (Zhu et al., 2019), m-Brick-Kiln* (Lee et al., 2021) (classification) and *m-Cashew-Plantation (Z. et al., 2021), m-SA-Crop-Type (Foundation, 2021)* (segmentation). Additional results on the remaining GEO-Bench datasets, including high-resolution tasks, are provided in Appendix C.3. **2) Copernicus-Bench** (Wang et al., 2025) provides 15 downstream tasks aligned with the full Sentinel mission family (Sentinel-1 to Sentinel-5P), and categorizes tasks into three levels: low-level (e.g., cloud detection), mid-level (e.g., land cover segmentation), and high-level (e.g., flood detection or yield prediction). While Copernicus-Bench leverages all Sentinel missions, in this work, we restrict evaluation to tasks using only Sentinel-1 and Sentinel-2 imagery. We evaluate on the following subset: *Cloud-S2 (Aybar et al., 2024), EuroSAT-S1 (Wang et al., 2024c), EuroSAT-S2 (Helber et al., 2019), BigEarthNet-S1 (Clasen et al., 2024), BigEarthNet-S2 (Clasen et al., 2024), DFC2020-S1 (Hänsch, 2019), DFC2020-S2 (Hänsch, 2019), LCZ-S2 (Zhu et al., 2019).*

## B  MULTI-MODAL FUSION STRATEGIES:

We investigate various strategies for multi-modal fusion and report results in Table A1 on two benchmark datasets: m-BigEarthNet and m-EuroSat. As a baseline, we evaluate standard DINO training using only Sentinel-2 L2A input (*DINO (S2-L2A)*), which learns unimodal representations. To enable explicit modality-aware learning, we apply a *Multi-Student-Teacher* approach where each modality has its own student and teacher networks, along with an alignment loss between student outputs to enforce cross-modal consistency. This yields consistent gains across both datasets. We also test a more expressive fusion approach, *CrossAttn (Q = 196) Global*, where 196 learned queries (standard for 224×224 image inputs) attend globally to multi-modal tokens immediately after patch embedding. However, this method does not perform well, likely due to excessive parameterization and lack of inductive bias for spatial alignment. Figure A1 visually summarizes key fusion strategies evaluated in Table A1, including (a) Multi-Student-Teacher, (b) unimodal DINO, and (c) CrossAttn (Q = 196) Global, highlighting their architectural differences and fusion mechanisms. Our proposed approach, *TerraFM-B (Q = 1)*, treats a modality as an augmentation and performs fusion using a single learned spatial query per location. This lightweight attention mechanism yields the best performance among non-ensemble methods. To further analyze architectural choices, we test a variant, *TerraFM-B (ViT PatchEmb)*, where the convolutional patch embedding is replaced by a ViT-S backbone purely for token extraction. While competitive, this setup slightly drops the performance due to increased model complexity and potential overfitting. Finally, our full model, *TerraFM-B (Q = 5)*, employs multiple learned spatial queries to achieve richer fusion between modalities. It achieves the best overall performance, validating the scalability and effectiveness of our fusion design.

|  | m-BigEarthNet | m-EuroSat |
|---|---|---|
| DINO (S2-L2A) | 54.6 | 83.2 |
| Multi-Student-Teacher | 55.8 | 87.8 |
| CrossAttn (Q = 196) Global | 52.0 | 77.1 |
| TerraFM-B (Q = 1) | 57.2 | 89.2 |
| TerraFM-B (ViT PatchEmb) | 56.9 | 87.2 |
| TerraFM-B (Q = 5) | **58.1** | **90.4** |

Table A1: Ablation study on multi-modal fusion strategies using k-NN evaluation. TerraFM-B with multiple spatial queries (Q = 5) achieves the best performance.

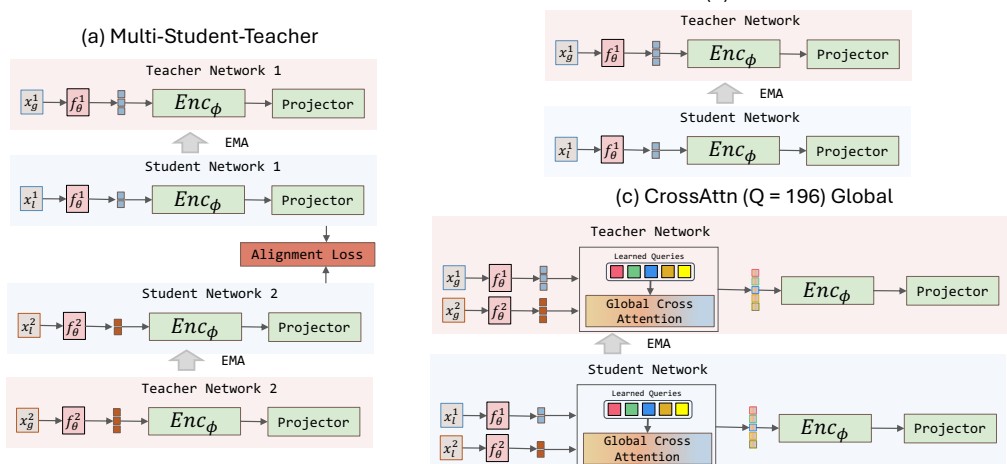

Figure A1: Architectural overview of different fusion strategies: (a) Multi-Student-Teacher with alignment loss, (b) unimodal DINO baseline, and (c) CrossAttn (Q = 196) with global learned queries.

## C  EVALUATION

### C.1  EVALUATION IMPLEMENTATION DETAILS

**Linear Probing Evaluation:** To evaluate the quality of learned representations, we follow a linear probing protocol of DINOv2(Oquab et al., 2023) that follows with a lightweight grid search over three key hyperparameters: (i) the learning rate, (ii) the number of transformer layers from which features are extracted, and (iii) whether to use only the [CLS] token or to concatenate it with the average-pooled patch tokens. We train the linear classifier using stochastic gradient descent (SGD) for 50 epochs. The training data is augmented using random resized cropping. Specifically, we sweep the learning rate over the set $\{1, 3, 4, 5\} \times 10^{\{-4, -3, -2, -1\}}$ Importantly, this search is computationally efficient: features from the frozen backbone are computed once per image using a single forward pass and reused across all configurations, since each linear head only requires a simple forward pass. For each configuration, we evaluate the classifier on the validation set and report the test accuracy achieved by the best validation configuration. **UperNet Probing Evaluation:** For UperNet (Xiao et al., 2018) Probing evaluation, we freeze the pretrained backbone and attach `UPerNet` decoder head. Specifically, we use a `Feature2Pyramid` module as the neck, followed by a `UPerNet` decoder and an auxiliary `FCNHead`. We train only the segmentation heads using the AdamW optimizer for 50 epochs without learning rate warm-up. We conduct a grid search over base learning rates $\{10^{-1}, 10^{-2}, 10^{-3}, 10^{-4}, 10^{-5}, 10^{-6}\}$. and batch size set $\{16, 32, 64\}$. **k-NN Evaluation:** To assess the quality of the learned representations without any finetuning, we apply non-parametric classification using a $k$-nearest neighbors (k-NN) classifier on the frozen features. In addition to sweeping over $k \in 3, 5, 7, 10, 15, 20, 30, 50, 100$ using validation set performance, we follow the same layer selection strategy as linear probing i.e evaluating features from the last 4 transformer layers. This protocol does not require additional training or data augmentation, making it a lightweight and reliable indicator of raw feature quality in pretrained models. **Finetuning Evaluation:** For full-model finetuning, we unfreeze the backbone and jointly optimize it with the task-specific head. We perform a grid search over learning rates in the evaluation set and batch

Table A2: Scaling behavior of TerraFM models with increasing model size and pretraining data across four GEO-Bench classification tasks.

| Dataset | Model | 20% | 100% | Gain |
|---------|-------|-----|------|------|
| EuroSat | TerraFM-S | 91.7 | 92.0 | 0.3 |
|         | TerraFM-B | 92.0 | 94.2 | 2.2 |
|         | TerraFM-L | 92.1 | 95.1 | 3.0 |
| BigEarthNet | TerraFM-S | 62.6 | 65.3 | 2.7 |
|             | TerraFM-B | 63.2 | 68.7 | 5.5 |
|             | TerraFM-L | 62.6 | 69.4 | 6.8 |
| So2sat | TerraFM-S | 50.5 | 52.3 | 1.8 |
|        | TerraFM-B | 49.7 | 55.1 | 5.4 |
|        | TerraFM-L | 49.1 | 55.9 | 6.8 |
| Brick-Kiln | TerraFM-S | 90.5 | 91.4 | 0.9 |
|            | TerraFM-B | 91.4 | 94.5 | 3.1 |
|            | TerraFM-L | 91.0 | 93.0 | 2.0 |

sizes. To stabilize training, we apply a reduced learning rate for the backbone, set to half of the main learning rate used for the head parameters. Once the best configuration is selected based on validation performance, we evaluate the finetuned model on the test set.

## C.2 Scaling Trends with Dataset Size:

We report scaling results on four GEO-Bench classification tasks when increasing model size and the pretraining dataset from 20% to 100% (Table A2). While all model sizes improve with additional data, the effect is more pronounced for the Base and Large variants. For example, TerraFM-L achieves a 6.8 point gain on BigEarthNet and So2Sat, compared to only 2.7 and 1.8 for TerraFM-S. On EuroSat and Brick-Kiln, where performance is already near saturation, the gains are smaller but still positive. These results confirm that larger models are more data-efficient and benefit disproportionately from increased pretraining scale, aligning with scaling laws observed in recent foundation model studies.

## C.3 Evaluation on High-Resolution Benchmarks.

To further assess generalization, we extend TerraFM's evaluation to include low-to-high resolution GEO-Bench tasks as well as the widely used AID (Xia et al., 2017) dataset (Table A3). Despite being pretrained solely on Sentinel-1 and Sentinel-2, TerraFM achieves consistent improvements over Galileo across diverse sensors and resolutions. Notably, TerraFM transfers effectively to m-forestnet, which uses 15m Landsat-8 inputs compared to TerraFM's 10m Sentinel-2 pretraining resolution, yielding a +7.7 point gain from baseline. On fine-scale RGB datasets such as m-pv4ger (0.1m) and m-chesapeake-landcover (1m), TerraFM also shows strong gains (+1.5 and +36.8 mIoU, respectively). These results highlight TerraFM's robustness across modalities and scales ranging from 0.1–15 m, complementing the evaluations in the main paper.

Table A3: Comparison on low-to high resolution benchmarks.

| Dataset | Task | Sensor | Resolution | Galileo | TerraFM |
|---------|------|--------|------------|---------|---------|
| m-forestnet (Irvin et al., 2020) | Classification | Landsat 8 | 15m | 49.4 | **57.1** |
| m-pv4ger (Mayer et al., 2022) | Classification | RGB | 0.1m | 96.7 | **98.2** |
| AID (Xia et al., 2017) | Classification | RGB | — | 78.2 | **93.8** |
| m-pv4ger-seg (Mayer et al., 2022) | Segmentation | RGB | 0.1m | 55.8 | **85.6** |
| m-chesapeake-landcover (Schmitt et al., 2019) | Segmentation | RGB | 1.0m | 14.6 | **51.4** |
| m-nz-cattle (Laradji et al., 2020) | Segmentation | RGB | 0.1m | 49.7 | **68.5** |
| m-NeonTree (Weinstein et al., 2020) | Segmentation | RGB | 0.1m | 51.1 | **54.0** |

## C.4 EVALUATION ON CHANGE DETECTION.

We evaluate TerraFM on the OSCD (Daudt et al., 2018) change detection dataset to assess the effect of sensor-invariance on temporal tasks (Table A4). Despite being trained without explicit temporal supervision, TerraFM-B with U-Net probing achieves 52.2 mIoU, substantially outperforming SSL4EO-S12 (Wang et al., 2022) (35.1), SEN12MS (Schmitt et al., 2019) (30.6), and SeCo (Mañas et al., 2021) (28.3). This suggests that TerraFM not only learns robust cross-sensor invariances but also implicitly learns time-invariant representations. However, due to the nature of DINO loss, which aligns global semantics, the model may still preserve object-level distinctions, resulting in improved performance on OSCD.

Table A4: Performance comparison on the change detection.

| Method | SeCo | SEN12MS | SSL4EO-S12 | TerraFM-B |
|---|---|---|---|---|
| F1 Score (%) | 28.33 | 30.62 | 35.08 | **52.20** |

# D ADDITIONAL ANALYSIS

## D.1 QUALITATIVE RESULTS:

Figure A2 illustrates qualitative results for the cloud and cloud shadow segmentation task from Copernicus-Bench. TerraFM accurately outlines both cloud and shadow regions, effectively distinguishing visually similar patterns while maintaining spatial coherence across varied scenes.

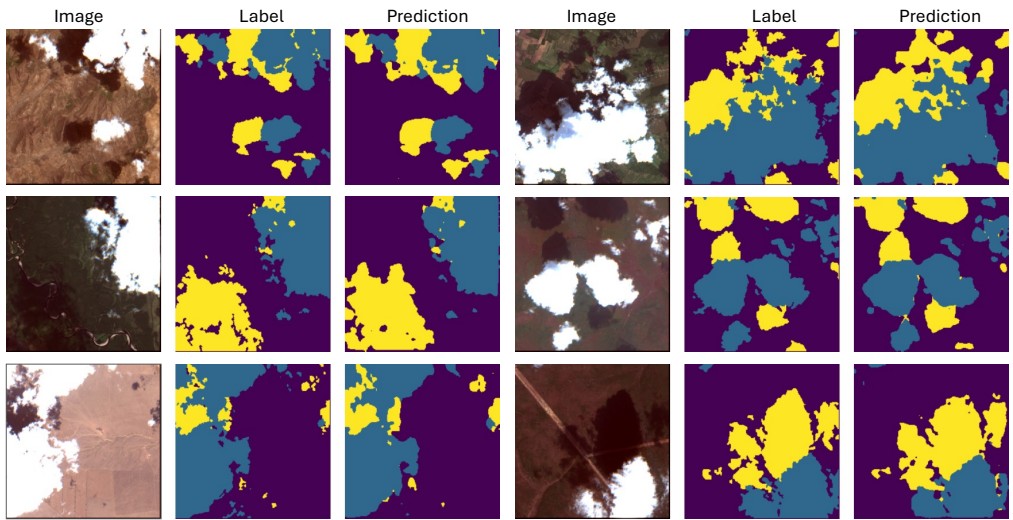

Figure A2: Qualitative results for cloud and cloud shadow segmentation. Each triplet shows the input image (left), the ground truth mask (middle), and the TerraFM prediction (right).

## D.2 GPU HOUR COMPARISON:

Compared to Prithvi-EO-v2.0, which trains ViT-L (300M) model using up to 80 GPUs for 400 epochs, consuming approximately 21,000 GPU-hours (Szwarcman et al., 2024), our TerraFM (300M) achieves comparable scale using significantly fewer resources. Specifically, TerraFM is trained for 200 epochs on 64 GPUs, amounting to approximately 12,000 GPU-hours.

### D.3 LANDSLIDE DETECTION

We evaluate landslide segmentation on the Landslide4Sense (L4S) (Ghorbanzadeh et al., 2022) benchmark, which provides segmentation labels for landslide and non-landslide regions across diverse mountainous areas using multi-source satellite data, including Sentinel-2 bands, DEM, and slope information. Our method, TerraFM, achieves strong performance with a mean IoU of 70.8 and a landslide IoU of 43.1, outperforming the Prithvi-EO-2.0 baseline (Table A5). Both TerraFM and Prithvi-EO-2.0 are trained using focal loss with a batch size of 16, Adam optimizer with a learning rate of $1 \times 10^{-4}$. Figure A3 shows qualitative results from TerraFM, illustrating predicted landslide masks alongside the ground truth. To assess variability, we repeated the Landslide4Sense experiment with three random seeds and observed stable results: TerraFM-B achieved $70.8 \pm 0.7$ mIoU and $43.1 \pm 0.9$ landslide IoU.

| | mIoU | IoU Landslide |
|---|---|---|
| Prithvi-EO-2.0 (300M) | 65.0 | 31.5 |
| TerraFM (120M) | **70.8** | **43.1** |

Table A5: Landslide detection performance on the Landslide4Sense test set. Despite being significantly smaller (120M parameters vs. 300M for Prithvi-EO-2.0), TerraFM achieves higher overall segmentation performance, especially for landslide regions.

### D.4 LAND COVER DISTRIBUTION:

Figure A5 illustrates the global spatial coverage of our pretraining data. The selected samples span diverse ecosystems, capturing a balanced mix of urban, vegetation, sea, and arid regions. The insets demonstrate fine-grained land cover variability, ensuring semantic richness across training tiles. This diverse geographic grounding plays a crucial role in enabling the generalization capabilities of TerraFM across regions and tasks.

**LLM Usage Statement:** We used large language models for polishing and improving the clarity of writing. They were not involved in research ideation, experiment design, analysis, or generation of results.

### D.5 PSEUDOCODE OF CROSS-ATTENTION FUSION

We summarize the cross-attention fusion mechanism used in TerraFM. The fusion module employs a small, fixed set of shared learnable queries (we use $N_q = 5$) that are applied uniformly across all spatial positions. For a $224 \times 224$ crop with a $16 \times 16$ patch size, each modality produces $N = 196$ spatial tokens. At each spatial location $n$, the modality-specific tokens at that position ($M$ tokens in total) serve as input to keys and values computation, while the shared queries attend over these modality tokens to produce $N_q$ intermediate outputs. These per-location outputs are then aggregated into a single fused token, yielding a fused sequence of length $N$ (plus the class token) for the ViT encoder. Thus, the fusion block preserves the backbone's original sequence length and performs modality mixing independently at each spatial location, with spatial interactions handled by subsequent transformer layers. Below, we provide a concise PyTorch-style pseudocode implementation of this per-location fusion mechanism.

```python
def cross_attention_fusion(Z_all, q, W_q, W_k, W_v, p_r, mha):

    # Z_all : [N, M, D]    tokens at N spatial positions, M modalities
    # q     : [N_q, D]     shared learnable queries (reused for all
        positions)
    # p_r   : [D, 1]       projection for scoring aggregated query outputs
    N, M, D = Z_all.shape
    N_q = q.shape[0]

    # Project shared queries once (reused across all spatial positions)
    Q = W_q(q)  # [N_q, D]

    fused_tokens = []
    for n in range(N):
        # Tokens from all modalities at spatial position n
        x_n = Z_all[n]             # [M, D]

        # Linear projections to keys and values
        K_n = W_k(x_n)             # [M, D]
        V_n = W_v(x_n)             # [M, D]

        # Cross-attention over modalities at position n
        # (Q attends to K_n, V_n; returns N_q outputs)
        z_prime_n, _ = mha(Q, K_n, V_n)  # [N_q, D]

        # Learned weighted mean over N_q query outputs
        scores = (z_prime_n @ p_r).squeeze(-1)  # [N_q]
        w = scores.softmax(dim=0)                # [N_q]
        z_fused_n = (w[:, None] * z_prime_n).sum(dim=0)  # [D]

        fused_tokens.append(z_fused_n)

    # Final fused sequence fed to the shared encoder Enc_phi
    Z_fused = torch.stack(fused_tokens, dim=0)  # [N, D]
    return Z_fused
```

Image            Ground Truth            TerraFM

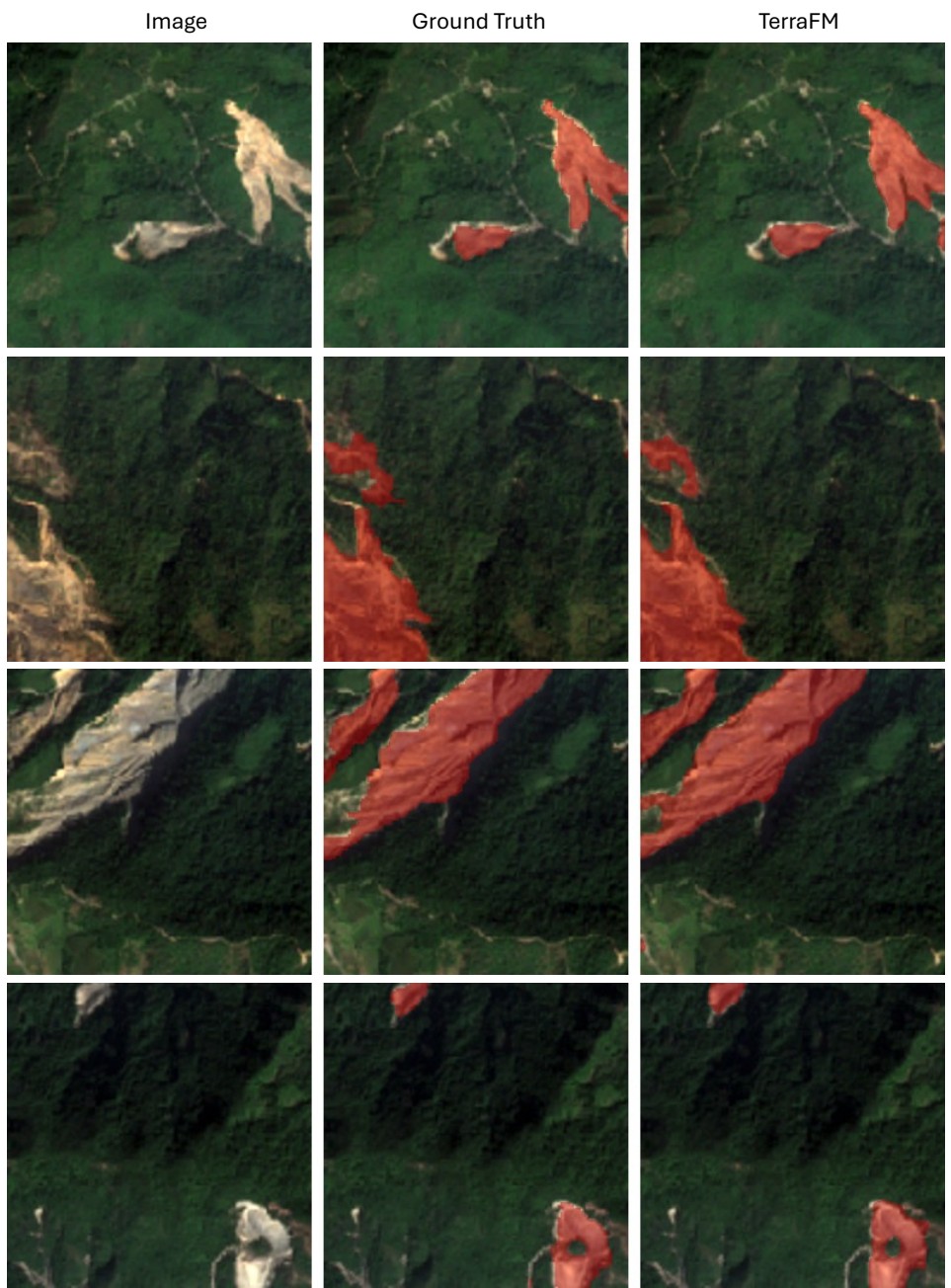

Figure A3: Qualitative results for landslide segmentation. Each triplet shows the input image (left), the ground truth mask (middle), and the TerraFM prediction (right).

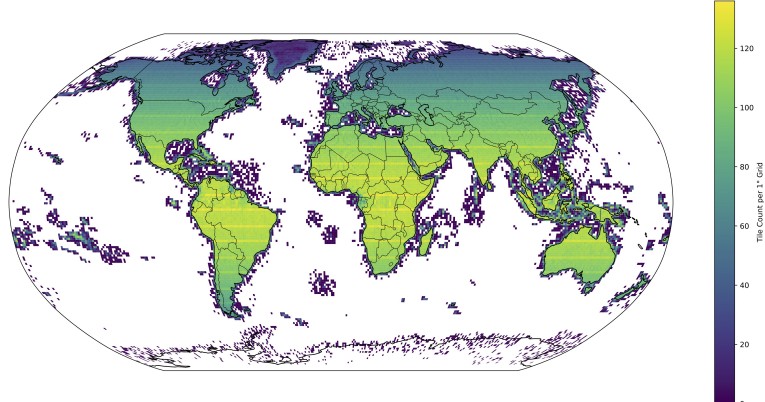

Figure A4: Global spatial distribution of the Major-TOM (Francis & Czerkawski, 2024) training subset. Each square shows a $1° \times 1°$ cell, colored by the number of 10.68 km × 10.68 km tiles it contains.

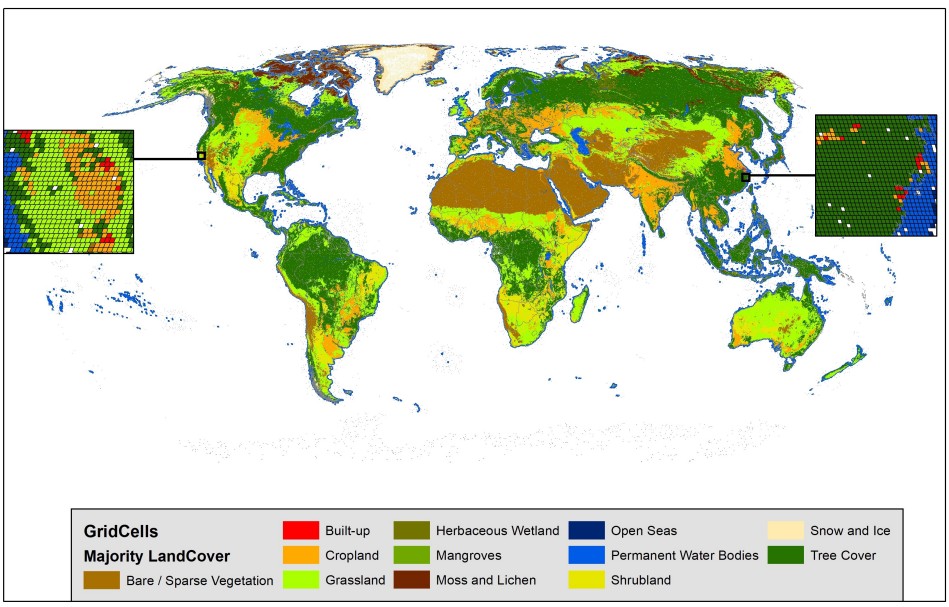

Figure A5: Global distribution of sampled training tiles by dominant land cover class, based on ESA WorldCover labels. Insets show detailed tile-level diversity, highlighting coverage across built-up, vegetation, and water classes.

