# OpenReview forum: "TerraFM: A Scalable Foundation Model for Unified Multisensor Earth Observation"
_ICLR.cc/2026/Conference — ICLR 2026 Poster_

### Official Review · Reviewer_Zhcb · 2025-10-25

**Soundness:** 2
**Presentation:** 1
**Contribution:** 2
**Rating:** 4
**Confidence:** 4

**Summary:**

The paper introduces TerraFM, which is a foundation model for remote sensing / earth observation. TerraFM leverages the following novel strategies: modality-specific patch embedding, natural augmentations, cross-attention fusion, and class-frequency-aware centering (in contrastive learning). TerraFM can process Sentinel-2 images (multispectral optical), Sentinel-1 images (synthetic aperture radar), or both. TerraFM achieves SOTA results across standard remote-sensing benchmarks.

**Strengths:**

- TerraFM achieves SOTA performance across standard RS benchmarks and over standard RS baseline methods
- The paper is, in general, clearly written and presented
- The cross-attention token-fusion strategy seems novel (although is unclear, see weaknesses)
- The class-frequency-aware centering strategy is novel to the best of my knowledge

**Weaknesses:**

- The novelty of the proposed strategies is often overstated and sometimes incorrect, in my view. For example, "modality-specific patch embedding" is just a different patch embedding layer per modality, which is commonly used (e.g., Galileo or even SatMAE). Treating sensor modalities as natural augmentations for contrastive learning is also not novel, e.g., CROMA performs radar-optical contrastive learning. Thus, the cross-attention fusion block and centering strategies are the only methods that I consider novel to TerraFM.
- The cross-attention fusion block is unclear to me. My understanding is that the images are patchified according to their modality, which become keys/values in cross-attention with learnable queries. However the number of learnable queries is unclear, as this determines the sequence length for the transformer backbone, this is crucial. In the appendix, it implies that there is 1 query per spatial location. Does this mean that with two modalities (2x196 patches), there are 196 learnable queries attending to 392 patches? The bottom of section 3.1 implies that each query only attends to the patches at the same spatial location. Can you please provide code to make this clear for me?

**Questions:**

- Please clarify the cross-attention fusion
- In Table 5, centering is ablated. Does this mean that both centering terms are removed together? If so, how can we know the effect of the novel centering term introduced?

---

> ### Author Response · Authors · 2025-11-23
> **Rebuttal Response**
>
> Dear Reviewer Zhcb,
>
> Thank you for your valuable feedback and constructive suggestions. Below, we provide detailed responses to each of the concerns raised.
>
> **Question 1: Addressing concerns on the novelty of TerraFM’s proposed components.**
>
> **Response 1:** We thank the reviewer for the observation. Please refer to our detailed clarification in `Reviewer omAQ Response 1`, where we explicitly outline which components are novel and how they differ from prior work.
>
> **Question 2: Clarification on cross-attention fusion block.**
>
> **Response 2:
> Number of queries and sequence length.**
> In TerraFM we do not use 196 distinct learnable queries. We use a small, fixed number of shared queries (we set $N_q = 5$) that are reused at all spatial locations. For an input of 224×224 with patch size 16, each modality yields ($N = 196$) tokens. At each spatial index $n$, we take only the $M$ modality-specific tokens at that location as input to keys and values computation (sequence length $M$), apply the shared $N_q$ queries, and then collapse them to a single fused token. The ViT backbone therefore, still operates on $N = 196$ fused tokens (plus class token), not on $2\times196$ tokens.
>
> **Per-location attention.**
> Yes, each query only attends to tokens from different modalities at the same spatial position; spatial mixing across locations is handled by the subsequent transformer blocks.
>
> In the revision, we have explicitly stated that queries are shared across locations and that the backbone’s sequence length remains $N$, and added a short, self-contained pseudo-code snippet in Appendix D5 to remove any ambiguity.
>
> **Question 3: Discussion on dual centering ablation.**
>
> **Response 3:** Thank you for pointing this out. In Table 5, the ablation labeled DC refers specifically to the dual-centering component introduced in TerraFM. The standard DINO centering term remains active in all rows, as it is an integral part of the teacher–student framework and cannot be removed without fundamentally changing the training objective.
>
> Thus:
>
> -   DC = off means: *DINO centering ON, dual-centering OFF*.
>
> -   DC = on means: *DINO centering ON, dual-centering ON*.
>
> This setup isolates the contribution of the new dual-centering term while keeping the baseline DINO mechanism intact. We have clarified this in the revised text.

---

### Official Review · Reviewer_Npar · 2025-10-30

**Soundness:** 3
**Presentation:** 3
**Contribution:** 3
**Rating:** 6
**Confidence:** 3

**Summary:**

The paper introduces TerraFM, a large-scale self-supervised foundation model for Earth Observation (EO) that unifies multi-sensor data—specifically Sentinel-1 SAR and Sentinel-2 optical imagery (L1C, L2A)—under a common framework. TerraFM is built on a student-teacher contrastive learning framework (similar to DINO) and introduces three main technical contributions to address challenges specific to EO data.The authors conduct extensive experiments on the GEO-Bench and Copernicus-Bench benchmarks. The results show that TerraFM consistently outperforms prior state-of-the-art models in both classification and segmentation tasks

**Strengths:**

1. Well-written and organized; clear motivation and architectural diagrams.
2. Strong empirical results: Their method outperforms all baselines on GEO-Bench and Copernicus-Bench for both classification and segmentation tasks.
3. Clear ablation studies isolate the impact of each component—modality augmentation, fusion, and dual centering—showing substantial performance gains

**Weaknesses:**

1. Although Major-TOM sampling is described, the paper gives limited discussion of potential geographic or seasonal biases in the curated dataset and how these might affect global transferability.
2. The model is designed for three specific inputs: S1, S2-L1C, and S2-L2A. While it generalizes well (Appendix C.3), the paper doesn't discuss the practical steps needed to extend TerraFM to a new modality (e.g., Landsat, hyperspectral).
3. The paper margin is smaller than the official format provided.
4. The cross-attention fusion module is activated "When multiple modalities M ⊆ M are selected" (L235-236). Given the stochastic assignment to student and teacher networks, this needs clarification. Does this mean fusion occurs within the student network if it happens to be assigned both S1 and S2 while the teacher does not use fusion?
5. The paper claims an advantage from using "large spatial tiles" (534px) to capture broader spatial context. However, the model's actual input resolutions are standard (224×224 for global crops, 96×96 for local crops). it is unclear to a reader that how much of the performance gain comes from this cropping strategy.

**Questions:**

1. How does the fusion module handle missing modalities at inference time? For example, if Sentinel-1 or Sentinel-2 data are unavailable, is there a fallback strategy?
2. Can the proposed modality-specific patch embedding extend easily to more than three modalities (e.g., DEM, hyperspectral data)?
3. Does TerraFM consider temporal alignment between Sentinel acquisitions, or is it purely spatial? If not, how might the framework extend to temporal FMs?
4. How sensitive is the model to the choice of the α parameter (balancing global and frequency-aware centers)? Was this tuned per dataset?

---

> ### Author Response · Authors · 2025-11-23
> **Rebuttal Response**
>
> Dear Reviewer Npar,
>
> Thank you for your valuable feedback and constructive suggestions. Below, we provide detailed responses to each of the concerns raised.
>
> **Question 1: Discussion of potential geographic or seasonal biases in data sampling.**
>
> **Response 1:** Thank you for pointing this out. Major-TOM samples at most one tile per grid cell, and, as illustrated in Fig. Ap4/Ap5, the resulting dataset is broadly distributed across continents and climate zones without strong regional dominance. Each cell independently selects a random 4-month window before cloud screening, which helps limit systematic seasonal bias. We have clarified these points in the revised manuscript.
>
> **Question 2: Discussion on how TerraFM handles missing modalities and can be extended to new modalities beyond S1/S2.**
>
> **Response 2:** Thank you for the question and appreciating the generalization aspect of our work. TerraFM can be extended to additional modalities in two practical ways. For downstream datasets that share only some channels with Sentinel-1/2, the missing bands can simply be zero-filled during fine-tuning, which is supported by our modality-specific patch embedding design (Appendix Ap3). For genuinely new modalities with no spectral overlap, such as thermal or hyperspectral, a new patch-embedding layer would need to be learned from co-located examples, after which the shared backbone can be used as is.
>
> **Question 3: Addressing formatting issue.**
>
> **Response 3:** Thank you for pointing this out. The margin issue was an unintentional formatting mistake during submission, and we sincerely apologize for this oversight. We have corrected the margins to fully comply with the official guidelines and adjusted the layout accordingly in the updated version to stay within the allowed space. We appreciate the reviewer bringing this to our attention.
>
> **Question 4: Clarification when and where cross-attention fusion is applied.**
>
> **Response 4:** In TerraFM, the fusion block is active in both the student and the teacher whenever a branch receives more than one modality. The stochastic view assignment simply determines which modalities go to each branch; if a branch happens to contain multiple modalities, fusion is applied in that branch.
>
> **Question 5: Clarification on advantage of using large 534px tiles during sampling**
>
> **Response 5:** Thank you for highlighting this point. The advantage of using 534×534 tiles is that the global and local crops are sampled from a much larger spatial field, even though the encoder ultimately receives 224×224 and 96×96 resized inputs. This preserves broader spatial context and more realistic neighborhood structure than cropping directly from small patches.
>
> **Question 6: Discussion on temporal alignment and extension to temporal foundation models.**
>
> **Response 6:** TerraFM is currently spatial-only because Major-TOM provides mono-temporal tiles, so the pretraining does not use any temporal alignment. If multi-date imagery is available, it can be incorporated without changing the backbone: each time step can go through the same modality-specific patch embedding and be used either as an additional view or fused within the modality before the existing cross-modal fusion block. As long as the temporal frames are spatially aligned or resampled, just as we do for S1/S2, this extension fits naturally into our framework.
>
> **Question 7: Sensitivity analysis on α hyperparameter.**
>
> **Response 7:** Thank you for highlighting this point. The α parameter is used only during pretraining as part of the dual-centering regularizer. We explored a small set of values during early development (α ∈ {1.0, 0.8, 0.5, 0}) to verify stability and ensure the regularizer behaved as intended. The trend was clear: α = 0.8 provided the strongest representation quality (e.g., 90.4% kNN on m-EuroSAT), while α = 0 eliminates the global centering term and forces all logits to be centered only by the head-class mean, creating a trivial uniform-softmax fixed point that caused the DINO loss to collapse. Based on this, we fixed α = 0.8 for the full pretraining run. Since dual-centering functions as a mild auxiliary regularizer and downstream tasks never revisit α, no per-dataset tuning is required.
>
> | **α** | **m-EuroSAT kNN OA (%)** |
> |-------|------------------------|
> | 1.0   | 88.50                  |
> | 0.8   | 90.40                  |
> | 0.5   | 88.80               |
> | 0.0   | N/A (collapsed)        |

---

### Official Review · Reviewer_YoBk · 2025-11-01

**Soundness:** 3
**Presentation:** 2
**Contribution:** 3
**Rating:** 8
**Confidence:** 3

**Summary:**

The authors introduce TerraFM, a S1/S2 foundation model for earth observation (EO) data. The paper proposes a modality-specific self-supervised pre-training recipe aimed specifically at EO domain-specific data characteristics. The authors incorporate a custom patch embedding mechanism catering to the multispectral nature of the data, treat different modalities as separate views for a multiview-learning setting, introduce an attention-based fusion mechanism to incorporate modalities within the model, and propose a dual-centering mechanism to handle the skewness of the underlying data distribution. Experiments indicate that the model robustly outperforms comparable foundation model efforts on two extensive benchmarking datasets.

**Strengths:**

- This paper has strong contributions to the literature, proposing a self-supervised pretraining strategy based on several well-motivated, domain-specific architecture components, including the double-centering strategy which is novel.
- The experiments are comprehensive and indicate strong performance, especially when including the supplement, building confidence in the quality of the model.

**Weaknesses:**

- The paper exceeds length requirements and ignores formatting guidelines (e.g. margins), this has to be fixed.
- The authors do not justify the choice not to incorporate any other augmentions in the pre-training framework. Many successful works  in the EO literature have previously done joint-embedding learning based on heavier augmentation strategies that are absent here, is there a reason?
- The main text (and appendix) lacks any details on model tuning or justification for hyperparameter choices.
- Table 2 and 3 use underlining to highlight second best models, this should be clear from the table description. Further, Table 4 does not use the same highlighting but should at least be consistent.
- Table 5 should show improvement from baseline SS performance for subsequent rows instead of the absolute scores, this would make it much more readable.

**Questions:**

- An complementary alternative explored to the dual-centering done to encourage learning long-tailed features of the data is altering dataset curation. While data sampling in this work is done based on land cover priors, have the authors considered replicating efforts like dynamic dataset curation (arXiv:2504.06962) instead or in tandem with the dual-centering strategy explored here to see how that impacts model performance?
- Similarly, since resolution varies between S1 and S2 imagery and acquisition modes, have the authors considered further augmenting the ViT backbone with domain-specific mechanisms, such as the scale information incorporated by ScaleMAE (arXiv:2212.14532)?

---

> ### Author Response · Authors · 2025-11-23
> **Rebuttal Response**
>
> Dear Reviewer YoBk,
>
> Thank you for your valuable feedback and constructive suggestions. Below, we provide detailed responses to each of the concerns raised.
>
> **Question 1: Addressing formatting issue.**
>
> **Response 1:** Thank you for pointing this out. The margin issue was an unintentional formatting mistake during submission, and we sincerely apologize for this oversight. We have corrected the margins to fully comply with the official guidelines and adjusted the layout accordingly in the updated version to stay within the allowed space. We appreciate the reviewer bringing this to our attention.
>
> **Question 2: Clarifying the decision to exclude hand-crafted augmentations in TerraFM.**
>
> **Response 2:** Thank you for highlighting this point. In TerraFM, we intentionally restrict augmentations to DINO-style multi-crop sampling. This choice is grounded in the properties of multisensor Sentinel-1/2 data: the combination of SAR backscatter, TOA reflectance, and surface reflectance already provides substantial natural variation in geometry, radiometry, and atmospheric conditions. As noted in our paper, the modalities themselves are treated as "natural augmentations" during training.
>
> In contrast, many heavy photometric or radiometric augmentations commonly used in RGB SSL do not translate well to EO data, particularly for S1/S2 pairs, because they can distort the physically meaningful cross-sensor relationships the model is meant to learn. For this reason, and consistent with prior multimodal EO work showing that handcrafted synthetic augmentations are less effective than naturally occurring cross-sensor variation ([Jain et al., 2022](https://sslneurips22.github.io/paper_pdfs/paper_34.pdf)), we focus on physically consistent augmentations. Empirically, this combination of multi-sensor diversity and multi-crop views proved sufficient and more stable for our training setup.
>
> **Question 3: Clarifying the justification for hyperparameter choices in TerraFM.**
>
> **Response 3:** Thank you for pointing this out. Most of the training hyperparameters in TerraFM, such as the learning-rate schedule, optimizer settings, crop configuration, and teacher momentum, are directly adopted from the standard DINO framework. We intentionally retained the DINO defaults to avoid method-specific tuning and to ensure fair comparability with prior self-supervised ViT work.
>
> **Question 4: Addressing table style consistency.**
>
> **Response 4:** Thank you for pointing out, we have updated the manuscript to make the highlighting explicit and consistent across Tables 2–4, and we now include a clear description of the underlining convention. Table 5 has also been revised to report the improvement over the baseline SS configuration alongside the absolute scores to improve readability.
>
>
> **Question 5: Considering dynamic dataset curation as an alternative or complementary strategy to dual centering.**
>
> **Response 5:** Thank you for the suggestion and for pointing out the connection to dynamic dataset curation. In TerraFM, our current pipeline already uses a simple, metadata-aware land-cover sampling strategy to ensure broad geographic and semantic coverage in the pretraining set. Dual-centering then operates at the representation level to rebalance long-tailed feature distributions during training.
>
> Dynamic dataset curation, such as the method proposed in arXiv:2504.06962, works at a different stage of the pipeline by selecting or pruning samples to increase dataset diversity and reduce redundancy. Because it modifies the data distribution itself rather than the feature distribution, we view it as complementary rather than overlapping with our approach. We have not yet combined the two strategies, but exploring joint data-level curation and representation-level balancing is a promising future direction, and we agree it may further benefit rare classes.
>
> **Question 6: Considering domain-specific scale mechanisms (ScaleMAE) to address S1–S2 resolution differences.**
>
> **Response 6:** Thank you for the suggestion. Although S2 bands have native resolutions of 10–60 m, all inputs in are harmonized to a 10 m grid and we did not introduce additional scale-specific modules. Despite this, TerraFM already shows strong cross-scale generalization: as reported in Table Ap3 , the model transfers well to datasets ranging from 0.1 m to 15 m resolution, including large gains on Landsat-8 (15 m) and fine-scale RGB segmentation (0.1–1 m). Incorporating explicit scale-aware mechanisms on top of this is a promising direction for future work.

---

### Official Review · Reviewer_omAQ · 2025-11-01

**Soundness:** 2
**Presentation:** 3
**Contribution:** 2
**Rating:** 6
**Confidence:** 5

**Summary:**

This paper presents TerraFM, a self-supervised remote sensing foundation model trained with Sentinel-2 and Sentinel-1 data. Compared to other remote sensing FMs, TerraFM has a much larger spatial input size. The model is trained using contrastive learning with different modalities used to construct augmentation pairs. The modalities are encoded separately and fused with cross-attention. The paper introduces a novel dual centering strategy that adds an additional prior about LULC distributions beyond the DINO centering strategy. Experiments on  Geo-Bench and Copernicus-Bench show that TerraFM outperforms most other models across most tasks.

**Strengths:**

- The paper is well-written and well-organized. It is easy to read and follow the technical details.
- The paper presents extensive experiments on Geo-Bench, Copernicus-Bench, and other datasets in the appendix. The ablation experiments test the contribution of each component.
- The performance is good across all tasks, with similar magnitudes across GeoBench tasks. This is particularly evident in Figure 1.

**Weaknesses:**

- The paper claims several novel contributions to the FM design: modality-specific patch embedding, modalities as augmentations, cross-attention fusion block, and dual-centering strategy. I think the dual-centering strategy is new (although very similar to the DINO centering), but it seems that all of the others have been proposed in prior work. The paper does not discuss how their contributions compare to these prior works.
    - modality-specific patch embedding: This is a common strategy in remote sensing FMs today. Maybe the authors realize this since it was not part of the ablation. I don’t think it can be claimed as a contribution. For example, SatMAE, Galileo, RingMo, and more all use separate embedding layers/tokenizers for each modality (or groups of channels within modalities). This was also done in vision with [MultiMAE](https://arxiv.org/pdf/2204.01678).
    - modalities as augmentations: The idea of using colocated pairs of images from different modalities as augmentation pairs for contrastive learning is also well established in prior work (e.g., [Jain et al., 2022](https://arxiv.org/abs/2209.02329); [Prexl & Schmitt 2023](https://openaccess.thecvf.com/content/CVPR2023W/EarthVision/papers/Prexl_Multi-Modal_Multi-Objective_Contrastive_Learning_for_Sentinel-12_Imagery_CVPRW_2023_paper.pdf); and CROMA).
    - cross-attention fusion block: I believe the same thing was done in [CrossMAE](https://openaccess.thecvf.com/content/CVPR2024/papers/Guo_CrossMAE_Cross-Modality_Masked_Autoencoders_for_Region-Aware_Audio-Visual_Pre-Training_CVPR_2024_paper.pdf) and other remote sensing specific work like [Chan-To-Hing & Veeravalli 2024](https://arxiv.org/abs/2401.02764).
- There is no discussion of performance variability and stability across runs/seeds or samples. Were the experiments repeated for different random seeds and/or subsets of the test data to quantify the standard error of performance metrics?
- I have some questions about comparisons between TerraFM and previous models (taken from Galileo paper) on downstream tasks:
    - Tseng et al. 2025 (Section C) describes the hyperparameter sweep done for evaluating downstream tasks. From the TerraFM paper’s Section C with evaluation implementation details, it seems there are similarities but not the same protocols compared to Galileo. I would expect TerraFM to use the same protocol as Galileo if they are trying to compare directly to numbers from Galileo’s tables.
    - The paper says the authors used linear probing, UperNet probing, k-NN, and fine-tuning for downstream task evaluation. It seems that all segmentation tasks except the ablation analysis used linear probing, but the ablation used UperNet probing for cashew plant. Why was the protocol changed here? Why was the cashew plant task chosen here? From my experience the labels in this dataset are noisier and lower resolution (and very sensitive to scale) compared to other geobench tasks.
- Some formatting and writing issues
    - The margins are smaller than allowed. This gives the paper a lot more space than other authors have to work with on their submitted papers. I expected this to constitute a desk reject.
    - The in-text citations need to be changed to use \citet (bracketed)
    - The style of Table 4 is notably different than the prevoius tables
- Minor issues
    - Table 1 has several confusing or unexplained details: I was confused that “Scale” represents the number of training samples not the number of model parameters. I would also have expected to find the # parameters in this table. Why is Benchmarks a relevant column here? How is Pixels (~T) computed?
    - There are some missing or unclear details about the pretraining data. The paper says “global distributional priors” were used - what priors/where did these come from? The paper says they “enriced each grid cell with metadata” - what metadata exactly and how was this formatted/interpreted by the model?
    - The “Impact of Components” paragraph discusses gain % for individual components, but those components are not isolated. For example, the 14% gain for cashew plant is from all of the components added, not just dual centering. Can the authors add rows to this table that isolate the contribution of each component?
    - The caption of Figure 6 talks about GeoBench as a whole, but the results are only shown for m-EuroSat.

**Questions:**

- What are the differences between the paper’s contributions and the works I referenced in the Weaknesses question?
- Were the experiments repeated for different random seeds and/or subsets of the test data to quantify the standard error of performance metrics?
- What are the differences between the evaluation protocols for the results taken from the Galileo paper and those applied for TerraFM?
- Why was the segmentation protocol changed from linear probe to upernet probe for the ablation experiment?
- WorldCover is known to have highly variable accuracy across the world, with especially poor performance in Africa (see [Kerner et al., 2024](https://www.nature.com/articles/s41597-024-03306-z) for one analysis). Since this is used as a prior in the dual centering method, do you expect performance to be worse in regions where WorldCover has worse performance?
- Can the authors add rows to the ablations table that isolate the contribution of each component?

---

> ### Author Response · Authors · 2025-11-23
> **Response 1/n**
>
> Dear Reviewer omAQ,
>
> Thank you for your valuable feedback and constructive suggestions. Below, we provide detailed responses to each of the concerns raised.
>
> **Question 1: Clarifying what is new in TerraFM and how it differs from prior approaches.**
>
> **Response 1:** We thank the reviewer for this clarification and agree that modality-specific patch embeddings and the use of co-located multimodal pairs are well-established in prior work. We do not claim these as standalone contributions. Our novelty lies in how these components are brought together within a unified DINO-style multi-crop distillation framework, which differs substantially from existing multimodal approaches.
>
> TerraFM operates all modalities (S1, S2-L1C, S2-L2A) as interchangeable multi-crop views inside a single shared student-teacher backbone, whereas prior multimodal SSL systems typically rely on separate unimodal encoders or multi-branch teacher-student setups (Fig. Ap1(a)), that leads to higher parameter counts and weaker cross-modal coupling. In addition, while earlier methods use pairwise contrastive alignment, TerraFM extends multi-crop distillation to heterogeneous modalities, which enables supervision across all combinations of global and local crops. This produces a stronger and more comprehensive alignment signal. As shown in Table Ap1, both the multi-student-teacher (separate encoders) and contrastive-style cross-attention variants perform noticeably worse than our shared-backbone formulation, indicating that the integration of these elements within multi-crop distillation, rather than the elements individually, is what drives the performance gains. We have clarified this more explicitly in the paper.
>
> **Cross-attention fusion:** We thank the reviewer for raising this point. We agree that cross-attention has been used in multimodal MAE-style models such as FUS-MAE and CrossMAE. The difference in TerraFM lies in how cross-attention is used and what objective it serves for our problem specifically.
>
> In prior work, cross-attention appears inside decoder-side reconstruction pipelines, either for pixel reconstruction or cross-embedding reconstruction, and is trained under a masked-autoencoding objective. In contrast, TerraFM introduces a lightweight encoder-side fusion block whose output becomes an additional DINO augmented view, inside a single student-teacher backbone, rather than being decoded to pixels. Our ablations (Table Ap1, Fig. Ap1) compare separate-encoder setups and cross attention global fusion and show that only the encoder-side fusion integrated into DINO distillation improves transfer performance. We have clarified this distinction in the revised manuscript.
>
> **Question 2: Discussion on performance variability and stability.**
>
> **Response 2:** We thank the reviewer for raising this point. To assess variability, we repeated the Landslide4Sense segmentation experiment using three different random seeds. The results were stable across runs, with TerraFM-B achieving 70.8 $\pm$0.7 mIoU and 43.1 $\pm$0.9 IoU for the landslide class. The values reported in Table Ap5 correspond to the mean performance across these runs. For other downstream datasets, we follow the standard practice in EO foundation model evaluations and report single-run results due to computational cost. We have added this discussion to the revised manuscript.
>
> **Question 3: Clarifying evaluation protocols relative to Galileo.**
>
> **Response 3:** We appreciate the reviewer highlighting this. Our initial experiments followed the standard DINO linear-probe settings, which we adopted because TerraFM is trained within a DINO-style teacher-student framework. To ensure a strictly fair comparison with Galileo, we re-evaluated the relevant segmentation tasks using the hyperparameter sweep described in Tseng et al. (2025, Section C). Under the matched protocol on m-Cashew-Plant and m-SA-Crop-Type, TerraFM continues to outperform all the baseline models. These updated results are reflected in the revised manuscript.
>
> | Model     | m-Cashew-Plant | m-SA-Crop-Type |
> |-----------|----------------|----------------|
> | Galileo   | 33.0           | 30.1           |
> | TerraFM-B | 34.1           | 32.8           |
> | TerraFM-L | 37.0           | 34.6           |
>
> **Question 4: Segmentation protocol differences in the ablation setting.**
>
> **Response 4:** Thank you for pointing this out. In the main results we use linear probing for segmentation to stay consistent with the evaluation protocol used in Galileo. For the ablation study, we additionally include UPerNet probing on m-Cashew-Plant because a decoder-based head exposes architectural differences more clearly than linear probing, making it more suitable for analyzing the contribution of each component. To avoid any protocol mismatch, we also provide the corresponding linear-probe scores (CP ‡) for all ablation variants in Table 5.

---

> > ### Author Response · Authors · 2025-11-23
> > **Response 2/2**
> >
> > **Question 5: Some formatting and writing issues.**
> >
> > **Response 5:** Thank you for pointing this out. The margin issue was an unintentional formatting mistake during submission, and we sincerely apologize for this oversight. We have corrected the margins to fully comply with the official guidelines and adjusted the layout accordingly in the updated version to stay within the allowed space. We appreciate the reviewer bringing this to our attention. Also, we have corrected the in-text citations to use the appropriate style, and we have updated Table 4 to match the formatting of the other tables for consistency. These changes are included in the revised version.
> >
> > **Minor Issues:**
> >
> > **Question 6: Addressing ambiguities in Table 1.**
> >
> > **Response 6:** Pixels (\~T) is computed by multiplying the number of tiles in each modality-resolution group by the pixel area of each tile and summing across all Sentinel-1 and Sentinel-2 inputs. Each modality (S1, S2-L1C, S2-L2A) contains approximately 6.24 million tiles, giving 18.7 million modality-specific tiles in total. For Sentinel-2, the native product includes 4 bands at 10 m (534×534 px), 6 bands at 20 m (267×267 px), and 2 bands at 60 m (89×89 px), all covering the same footprint but differing in pixel density before resampling. Sentinel-1 RTC, aligned to the same 10 m grid, contributes two 534×534 px tiles per grid cell. The total pixel count used in Table 1 is therefore obtained by summing the contributions of all modalities and resolutions:
> >
> > Pixels (S2L1C+S2L2A+S1): ((534×534×4)+(267×267×6)+(89×89×2))×6.24M+((534×534×4)+(267×267×6)+(89×89×2))×6.24M+(534×534×2)×6.24M
> >
> > Moreover, we have clarified the "Scale" terminology in the caption.
> >
> > **Question 7:  Clarifying global priors and grid-cell metadata.**
> >
> > **Response 7:** By "global distributional priors", we refer to the per-grid land-cover label derived from ESA WorldCover. For each Major-TOM grid cell, we compute the majority land-cover class and store this as a single categorical value. This label is not provided to the model as input or used as supervision; it is only used to compute class-frequency statistics for the dual-centering regularizer. No other metadata is used. We have clarified this in the revised manuscript.
> >
> >
> >
> > **Question 8: Addressing concerns about WorldCover noise and regional bias.**
> >
> > **Response 8:** We agree that WorldCover has notable regional variability in accuracy, including in parts of Africa. In TerraFM, however, this information is used only at the tile level; each 534×534 grid cell is assigned a single majority land-cover label, which substantially smooths pixel-level noise and reduces the impact of local misclassification. Importantly, this label is not used as a supervision signal for training; it enters only as a soft prior in the dual-centering mechanism rather than a hard constraint. As a result, we do not expect substantial regional degradation tied to WorldCover noise.
> >
> >
> > **Remaining Minor Issues:**
> > Thank you for pointing out the remaining minor issues. All corrections are included in the revised version.

---

### Meta-Review · Area_Chair_MGMz · 2025-12-28

**Summary:**

All four reviewers found the paper well written and the empirical results strong. However, they raised concerns about the method’s novelty, the experimental setup, and generalizability. The rebuttal addresses some of these points, though not all. Overall, given that TerraFM achieves state-of-the-art performance on GEO-Bench and Copernicus-Bench, I view it as a solid contribution to multisensor Earth observation foundation models and recommend acceptance.

### Novelty

Several reviewers (**omAQ**, **Zhcb**) questioned the paper’s novelty. Reviewer **omAQ** noted that some claimed contributions—modality-specific patch embeddings, treating modalities as augmentations, and the cross-attention fusion block—have appeared in prior work, and reviewer **Zhcb** echoed the concern specifically about the patch embedding. Both reviewers, however, agreed that the class-frequency-aware centering strategy is novel.

In the rebuttal, the authors argue that TerraFM’s main novelty lies in adopting a single shared student–teacher backbone with interchangeable multi-crop views, whereas prior multi-sensor Earth observation methods typically rely on separate unimodal encoders or multi-branch teacher–student designs. They also attempt to justify the cross-attention fusion module as a contribution. I remain unconvinced on the cross-attentino fusion module: the fusion design appears closely aligned with the cross-attention mechanisms used in FUS-MAE and CrossMAE, and overall feels less novel than the propos

### Experimental Setting

Reviewers (**omAQ, YoBk, Npar, Zhcb**) also raised concerns about the experimental protocol and the fairness/clarity of baseline comparisons. Reviewer **omAQ** asked about performance variability, the hyperparameter sweep protocol, and the probing method used for evaluation; the authors addressed these points with updated results and additional details. Reviewer **YoBk** noted that the paper lacked information on model tuning and hyperparameter selection, which the authors clarified in their response. Reviewer **Npar** questioned the rationale for the 534px tile size, the design choice of multimodal fusion in the teacher model, and the sensitivity to the α parameter; these were answered in the rebuttal with new experiments. Finally, reviewer **Zhcb**’s question about the meaning of “DC” in Table 5 was also clarified in the rebuttal.

### Generalizability

Reviewer **Npar** questioned the method’s generalizability, specifically how it would extend to additional modalities and whether geographic or seasonal biases in Major-TOM could limit its applicability. In the rebuttal, the authors describe steps taken to mitigate these biases and outline practical approaches for incorporating new modalities.

**Reviewer Concerns:**

Overall, the rebuttal adequately addresses the concerns about the experimental setting and the method’s generalizability, while the novelty concerns are only partially resolved. TerraFM is meaningfully differentiated from prior multi-sensor Earth observation foundation models in its use of a single shared student–teacher backbone with multi-crop inputs. However, several of the claimed contributions—such as modality-specific patch embeddings and the cross-attention fusion block—appear to overlap with existing prior work.

**Reviewer Scores:**

I expect reviewers **omAQ** and **Npar** may raise their scores after the rebuttal, given that several of their concerns were addressed with added details and experiments. Reviewer **Zhcb**’s final score will likely hinge on whether they view the cross-attention fusion block as a central—and sufficiently novel—contribution of the paper.

---

### Decision · Program_Chairs · 2026-01-26

Accept (Poster)